# Precise transcriptional control of cellular quiescence by BRAVO/WOX5 complex in *Arabidopsis* roots

Isabel Betegón-Putze[1,†], Josep Mercadal[2,3,†], Nadja Bosch[1], Ainoa Planas-Riverola[1] (iD),
Mar Marquès-Bueno[1], Josep Vilarrasa-Blasi[1,§], David Frigola[2], Rebecca C Burkart[4],
Cristina Martínez[5], Ana Conesa[6], Rosangela Sozzani[7] (iD), Yvonne Stahl[4], Salomé Prat[5],
Marta Ibañes[2,3,**,‡] (iD) & Ana I Caño-Delgado[1,*,‡] (iD)

## Abstract

Understanding stem cell regulatory circuits is the next challenge in plant biology, as these cells are essential for tissue growth and organ regeneration in response to stress. In the Arabidopsis primary root apex, stem cell-specific transcription factors BRAVO and WOX5 co-localize in the quiescent centre (QC) cells, where they commonly repress cell division so that these cells can act as a reservoir to replenish surrounding stem cells, yet their molecular connection remains unknown. Genetic and biochemical analysis indicates that BRAVO and WOX5 form a transcription factor complex that modulates gene expression in the QC cells to preserve overall root growth and architecture. Furthermore, by using mathematical modelling we establish that BRAVO uses the WOX5/BRAVO complex to promote WOX5 activity in the stem cells. Our results unveil the importance of transcriptional regulatory circuits in plant stem cell development.

**Keywords** BRAVO; mathematical modelling; quiescent centre; root growth; WOX5

**Subject Categories** Chromatin, Transcription & Genomics; Development; Plant Biology

**Mol Syst Biol. (2021) 17: e9864**

## Introduction

Roots are indispensable organs to preserve plant life and terrestrial ecosystems under normal and adverse environmental conditions. In *Arabidopsis thaliana* (Arabidopsis), the primary root derives from the activity of the stem cells located at the base of the meristem in the root apex (Dolan *et al*, 1993; van den Berg *et al*, 1995). The root stem cell niche (SCN) is composed of a set of proliferative stem cells that surround the mitotically less active cells, named the quiescent centre (QC; Scheres, 2007). Proximally to the QC, the vascular stem cells (VSC, also called vascular initial cells) and the cortical endodermal initials give rise to functional procambial, xylem and phloem conductive vessels and the ground tissue, respectively. Distally and laterally to the QC, the columella stem cells (CSC) and the lateral root cap and epidermal initials give rise to the most outer layer root tissues (Appendix Fig S1; Stahl *et al*, 2009; Gonzalez-Garcia *et al*, 2011; De Rybel *et al*, 2016). The QC prevents differentiation of all these surrounding stem cells (van den Berg *et al*, 1997), and its low proliferation rate provides a way to preserve the genome from replication errors (Cheung & Rando, 2013). It also acts as a root stem cells reservoir, having the ability of promoting its own division rate to replenish the stem cells when they are damaged (Fulcher & Sablowski, 2009; Lozano-Elena *et al*, 2018).

BRASSINOSTEROIDS AT VASCULAR AND ORGANIZING CENTER (BRAVO) and WUSCHEL RELATED HOMEOBOX 5 (WOX5) are two transcription factors (TFs) that are expressed in the QC and control its quiescence, as mutation of either BRAVO or WOX5 promotes QC cell division (Forzani *et al*, 2014; Vilarrasa-Blasi *et al*, 2014; Pi *et al*,

1 Department of Molecular Genetics, Centre for Research in Agricultural Genomics (CRAG), CSIC-IRTA-UAB-UB, Campus UAB (Cerdanyola del Vallès), Barcelona, Spain
2 Departament de Matèria Condensada, Facultat de Física, Universitat de Barcelona, Barcelona, Spain
3 Universitat de Barcelona Institute of Complex Systems (UBICS), Barcelona, Spain
4 Institute for Developmental Genetics, Heinrich-Heine University, Düsseldorf, Germany
5 Department of Plant Molecular Genetics, Centro Nacional de Biotecnología (CNB), Madrid, Spain
6 Microbiology and Cell Science, Institute for Food and Agricultural Research, Genetics Institute, University of Florida, Gainesville, FL, USA
7 Department of Plant and Microbial Biology, North Carolina State University, Raleigh, NC, USA
*Corresponding author. Tel: +34 935636600 ext 3210; E-mail: ana.cano@cragenomica.es
**Corresponding author. E-mail: mibanes@ub.edu
†These authors contributed equally to this work as first authors
‡These authors contributed equally to this work as senior authors
§Present address: Department of Biology, Stanford University, Stanford, CA, USA

2015). BRAVO is an R2R3-MYB transcription factor and besides being expressed in the QC, it is also present at the vascular initials (Vilarrasa-Blasi *et al*, 2014). It was identified as a target of Brassinosteroid (BR) signalling, being directly repressed by BRI1-EMS-SUPPRESSOR 1 (BES1), one of the main effectors of the BR signalling pathway, altogether with its co-repressor TOPLESS (TPL; Vilarrasa-Blasi *et al*, 2014; Espinosa-Ruiz *et al*, 2017). WOX5 is a member of the WUSCHEL homeodomain transcription factor family, and it is localized mainly in the QC and to a lesser extent at the surrounding CSC and vascular initials (Sarkar *et al*, 2007; Pi *et al*, 2015). WOX5 can repress QC divisions by repressing CYCLIN-D3;3 (Forzani *et al*, 2014) and, in contrast to BRAVO, is also involved in CSC differentiation, as in the *wox5* mutant CSC differentiate prematurely (Sarkar *et al*, 2007).

Although BRAVO and WOX5 are well-studied plant cell-specific repressors of QC division, their molecular connection and the biological relevance in SCN proper functioning has not yet been established. In this study, we set the regulatory and molecular interactions between BRAVO and WOX5 at the SCN and disclose a common role as regulators of primary and lateral root development. Our results show that BRAVO and WOX5 promote each other's expressions and can directly bind to form a protein regulatory complex with common downstream regulators in the QC cells. BRAVO/WOX5 protein interaction underlies their functions as QC repressors to maintain stem cell development, that is essential for root growth.

## Results

### BRAVO and WOX5 control QC division and lateral root density

We have previously shown that *bravo* mutants have increased divisions at the QC compared to the wild type (WT; Vilarrasa-Blasi *et al*,

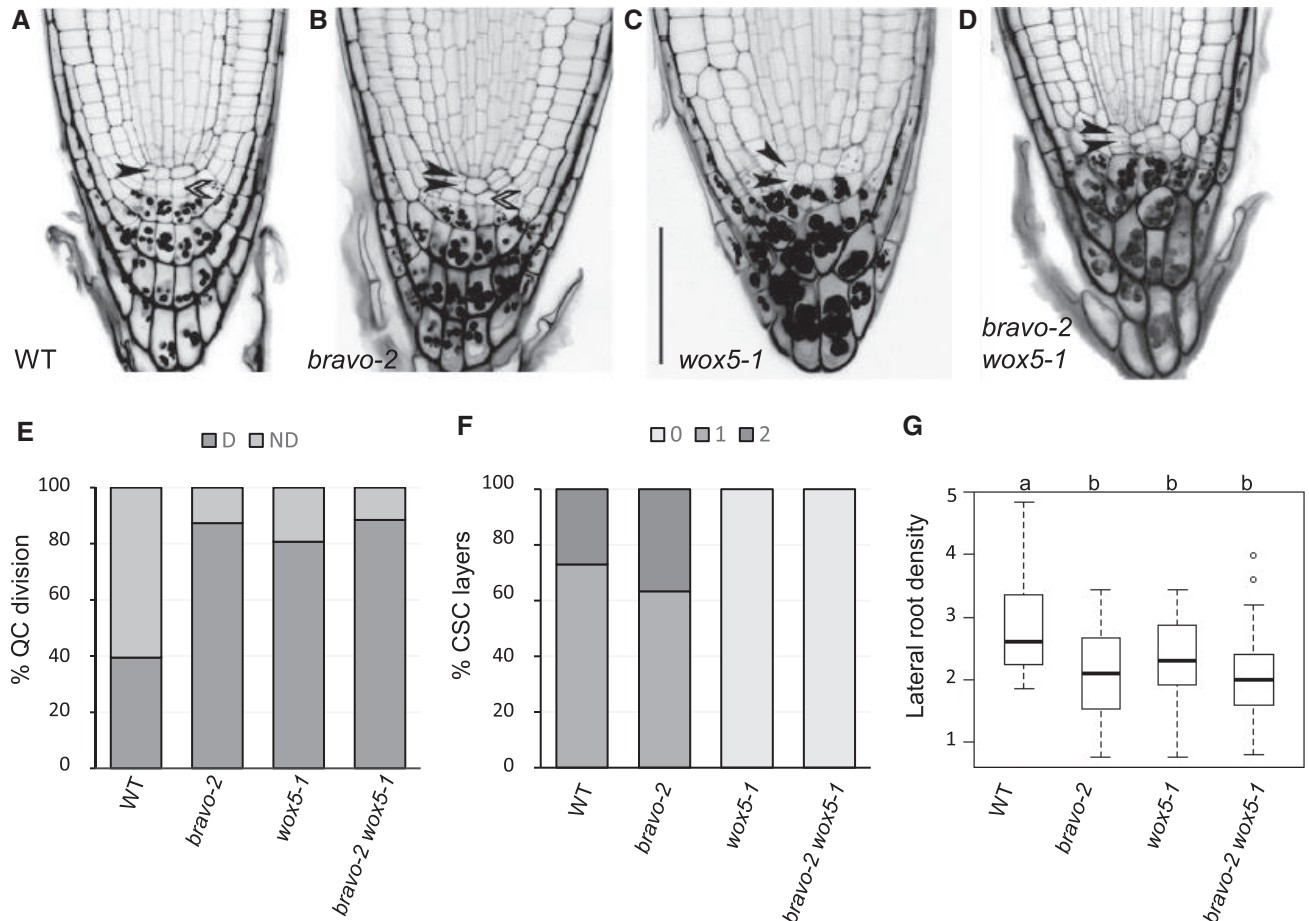

**Figure 1.  BRAVO and WOX5 are required for the QC identity and stem cells maintenance.**

A–D   Confocal images of mPS-PI-stained 6-day-old seedlings of Col-0 (A), *bravo-2* (B), *wox5-1* (C) and *bravo-2 wox5-1* (D) mutants. Left black arrows indicate QC cells, and right white arrows indicate CSC. Scale bar: 50 μm.

E   Quantification of the QC divisions in 6-day-old roots expressed in percentage (*n* > 50, 3 replicates). D: QC divided; ND: QC non-divided.

F   Quantification of CSC layers in 6-day-old roots expressed in percentage (*n* > 50, 3 replicates).

G   Lateral root density (number of lateral roots per mm of root length) of 7-day-olf WT, *bravo-2*, *wox5-1* and *bravo-2 wox5-1* mutants (*n* > 40, 2 replicates). Different letters indicate statistically significant differences (*P*-value < 0.05 Student's *t*-test). In the boxplot, box width represents the interquartile range (IQR = Q3-Q1), with the horizontal line denoting the median, while whiskers extend from Q1-1.5IQR to Q3+1.5IQR. White dots are the outliers.

2014; Fig 1A and B), which resemble the ones described for *wox5* mutant (Sarkar *et al,* 2007; Bennett *et al,* 2014; Forzani *et al,* 2014; Fig 1C). To address whether BRAVO and WOX5 together play a repressing function for QC divisions, we generated the double *bravo wox5* mutants (Materials and Methods, Appendix Table S1). The *bravo wox5* double mutants also exhibited increased cell division compared to the WT (Fig 1A and D). Importantly, the frequency of divided QC was similar to that of *bravo* and *wox5* single mutants (Fig 1E). The mutual epistatic effect of these mutations suggests that BRAVO and WOX5 function interdependently at the WT primary root apex to suppress QC divisions.

Previous studies proposed that WOX5 represses CSC differentiation in a non-cell autonomous manner (Sarkar *et al,* 2007; Bennett *et al,* 2014), whereas no link was reported between this process and BRAVO, since the *bravo* mutants are not defective in CSC differentiation (Fig 1A, B and F). Genetic analysis showed that *bravo wox5* mutants display the same CSC differentiation as *wox5* single mutant (Fig 1A, C, D and F), corroborating that BRAVO does not control CSC differentiation (Vilarrasa-Blasi *et al,* 2014).

To address whether these stem cell-specific defects account for overall alterations in root growth and development, root architecture was analysed. The *bravo wox5* double mutant shows slightly but significantly shorter roots than the WT (Appendix Fig S2A) and lower lateral root density (Fig 1G). In the case of the lateral root density, 7-day-old *bravo wox5* seedlings show the same phenotype as the single mutants (Fig 1G), in agreement with previous reports for *wox5* (Tian *et al,* 2014a). Root growth defects become more exaggerated in the *bravo wox5* double mutant than WT and single mutants in 10-day-old seedlings (Appendix Fig S2B), therefore supporting their joint contribution to overall root growth and architecture.

### *BRAVO* and *WOX5* control each other's expression at the root stem cell niche

We have previously shown that *WOX5* expression is reduced in the *bravo* mutant (Vilarrasa-Blasi *et al,* 2014), indicating that BRAVO regulates *WOX5* expression. To gain insight on the mutual regulatory activity of these two transcription factors, we investigated *BRAVO* and *WOX5* expressions at the SCN in the single mutant and in the double *bravo wox5* mutant backgrounds.

In the WT primary root, *BRAVO* expression, reported by the *pBRAVO:GFP* line, is specifically located in the QC and the vascular initials (Vilarrasa-Blasi *et al,* 2014; Fig 2A). The *pBRAVO* signal was increased in the *bravo* mutant extending shootwards (Fig 2B and H), suggesting that BRAVO negatively regulates its own expression. In contrast, in the *wox5* mutant, *pBRAVO* expression was strongly reduced, especially at the QC, suggesting that WOX5 promotes *BRAVO* expression (Fig 2C and H). Inducible expression of WOX5 under the 35S promoter (35S:WOX5-GR) resulted in an increased total *BRAVO* expression, as measured by RT–qPCR of root tips (Appendix Fig S3A). Together, these results support that WOX5 activates *BRAVO* expression. Moreover, *pBRAVO* expression was equally reduced in the double *bravo wox5* mutant (Appendix Fig S4), as in the *wox5* mutant (Fig 2C and H), suggesting that the BRAVO induction by WOX5 is stronger than the BRAVO self-repression.

In the primary root, *WOX5* expression, as reported by the *pWOX5:GFP* line, is enriched in the QC, with some expression detected in the vascular initials (Pi *et al,* 2015; Clark *et al,* 2020;

Fig 2D). We found that *bravo* mutant displayed a reduction of *WOX5* expression (Fig 2E and I; Vilarrasa-Blasi *et al,* 2014), supporting that BRAVO in turn is able to control the expression of the *WOX5* gene. Further analysis of *WOX5* expression upon overexpressing *BRAVO* under an inducible 35S promoter (35S:BRAVO-Ei) showed that when BRAVO levels were induced, *pWOX5* levels remained similar to the WT, indicating that BRAVO is not sufficient to induce an increase of *WOX5* expression (Appendix Fig S3C–G). Together, these results support that BRAVO is necessary to maintain proper *WOX5* levels in the QC. Subsequently, an increased *pWOX5: GFP* expression towards the provascular cells and columella stem cells was observed in the *bravo wox5* double mutant (Fig 2G), similar to *wox5* mutant (Fig 2F and I). These findings suggest that *WOX5* regulates its own expression and restricts it to the QC, while BRAVO helps to maintain *WOX5* expression.

### Two possible scenarios for BRAVO/WOX5 cross-regulations

To provide a comprehensive scheme of *BRAVO* and *WOX5* cross-regulation in the SCN able to account for the changes in expression levels observed in the various mutant backgrounds and overexpression lines, we turned to mathematical modelling. We formulated two models for these regulations, differing by the way *BRAVO* regulates the activity of *WOX5* (Fig 3A and B, Materials and Methods). In both cases, the modelling approach took as variables the total *BRAVO* and total *WOX5* expressions at the root tip (mimicking the total *pBRAVO:GFP* and *pWOX5:GFP* expressions integrated across the SCN; Fig 2H and I) and modelled effective cross-regulations between them that can drive all experimental observations on the promoter fold-changes between WT and mutants, and overexpressing lines. These effective regulations are expected to encompass several possible transcriptional and post-transcriptional mechanisms, which may take place across the whole SCN, within or between cells.

Because *BRAVO* is induced in the WOX5 overexpression line (Appendix Fig S3A) and *pBRAVO:GFP* expression decreases in the *wox5* mutant (Fig 2C), both models considered that *WOX5* expression induces (either directly and/or through intermediate molecules) the expression of *BRAVO* (Fig 3A and B). To account for the increased total *pBRAVO:GFP* expression in the *bravo* background (Fig 2B), the models assumed that *BRAVO* is able to inhibit its own total expression (Fig 3A and B), a regulation that is probably indirect. When the induction by *WOX5* is stronger than the *BRAVO* self-inhibition, these two regulations can account for a decrease in *BRAVO* expression in the *bravo wox5* double mutant, as found by the GFP expression data (Appendix Fig S4). Therefore, these two regulations on *BRAVO* are expected to be sufficient to account for all the changes of its expression that we observed in the single and double mutants as well as in the inducible overexpression of WOX5 with respect to WT.

Because *pWOX5:GFP* expression in the SCN increases in the *wox5* mutant (Fig 2F), both models consider that *WOX5* represses (directly and/or indirectly) its own promoter activity (Fig 3A and B). In contrast, two different regulations of *WOX5* by *BRAVO* were hypothesized to account for both the decreased *pWOX5:GFP* expression in the *bravo* mutant (Fig 2E and I) and the absence of change in *WOX5* expression upon BRAVO overexpression (Appendix Fig S3C–G). In one model, hereafter named "alleviation model", *BRAVO*

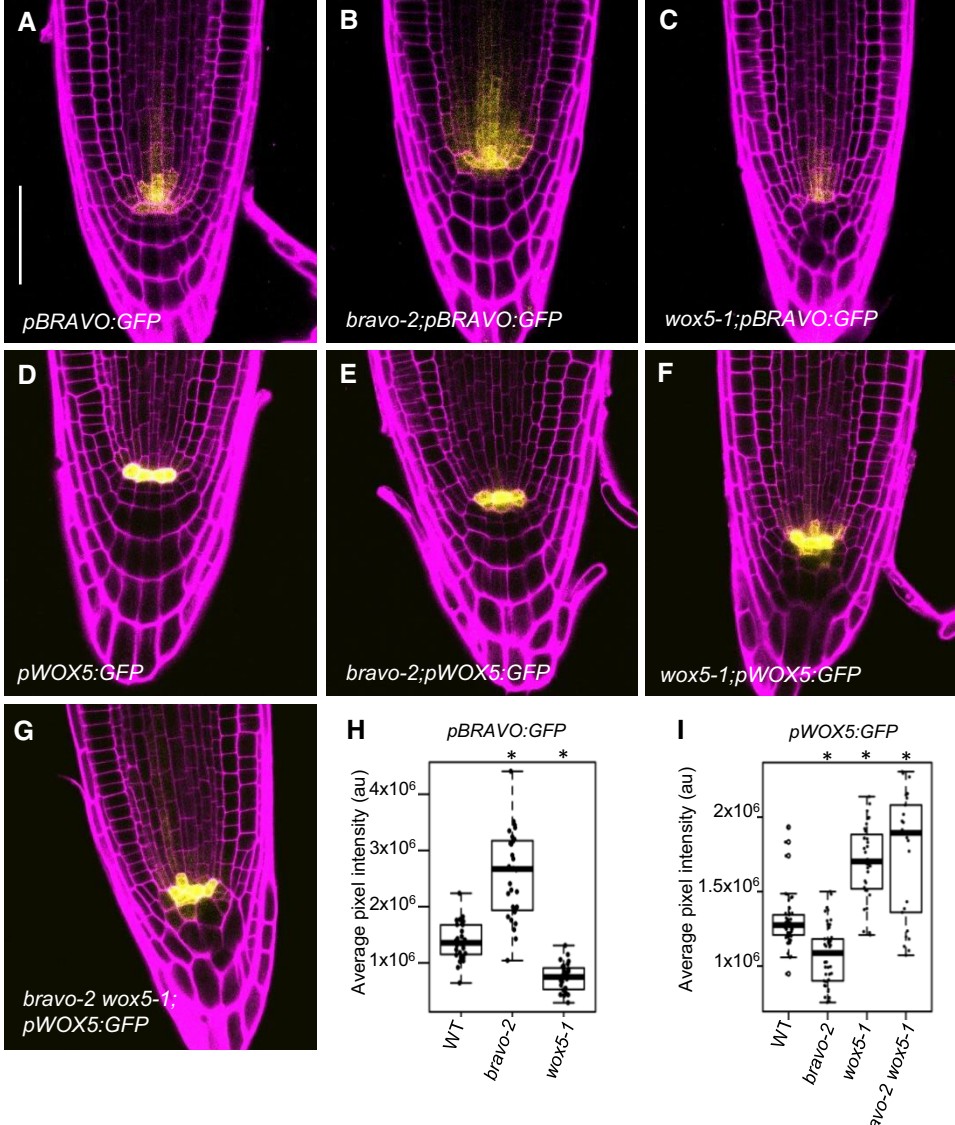

**Figure 2. BRAVO and WOX5 reinforce each other at the root stem cell niche.**

A–G Confocal images of PI-stained 6-day-old roots. GFP-tagged expression is shown in yellow. A-C) pBRAVO:GFP in WT (A), bravo-2 (B) and wox5-1 (C) knockout backgrounds. D-G) pWOX5:GFP in the WT (D), bravo-2 (E), wox5-1 (F) and bravo-2 wox5-1 (G) knockout backgrounds. Scale bar: 50 μm.

H, I Quantification of the GFP fluorescent signal of the roots in A-C (H) and D-G (I). Boxplot indicating the average pixel intensity of the GFP in the stem cell niche (n > 25, 3 biological replicates, *P-value < 0.05 Student's t-test for each genotype versus the WT in the same condition). Quantification was done by integrating the GFP signal in each root across defined areas that included the whole SCN (Appendix Fig S7). In the boxplot, box width represents the interquartile range (IQR = Q3-Q1), with the horizontal line denoting the median, while whiskers extend from Q1-1.5IQR to Q3+1.5IQR. White dots are the outliers, and black dots are the experimental observations.

inhibits partially WOX5 self-repression (Fig 3A). Therefore, this model proposes that BRAVO promotes WOX5 expression by alleviating WOX5 self-inhibition. In the other model, hereafter named "activation model", BRAVO activates WOX5 (Fig 3B).

With these interactions, the two models precisely capture all changes in BRAVO and WOX5 expression in the bravo, wox5 and bravo wox5 mutants as well as in the overexpressing lines (Appendix Fig S8). Parameter values (Appendix Table S1) were chosen such that the fold-changes between expression levels in the

single mutants compared to the WT matched the fold-changes in the GFP and transcript levels of our empirical data in mutants and over-expressing lines (Appendix Fig S8, Materials and Methods). Nonetheless, the GFP fold-change values have to be taken as qualitative trends rather than specific quantitative fold-changes in mRNA levels. In addition, the parameter values were restricted such that under control conditions BRAVO expression is lower than WOX5 expression in the WT (Appendix Fig S8A and B), as suggested by pBRAVO:GFP and pWOX5:GFP GFP expression (Materials and

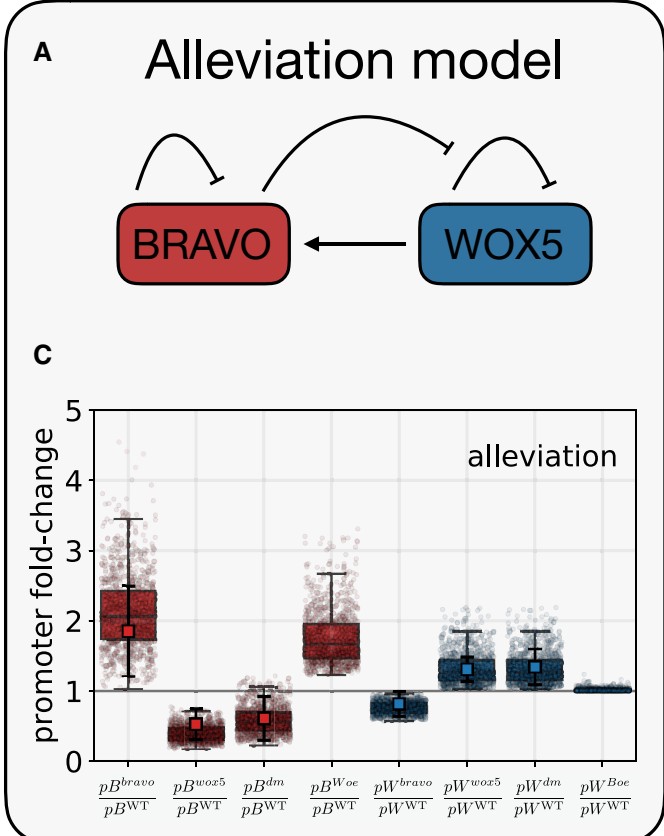

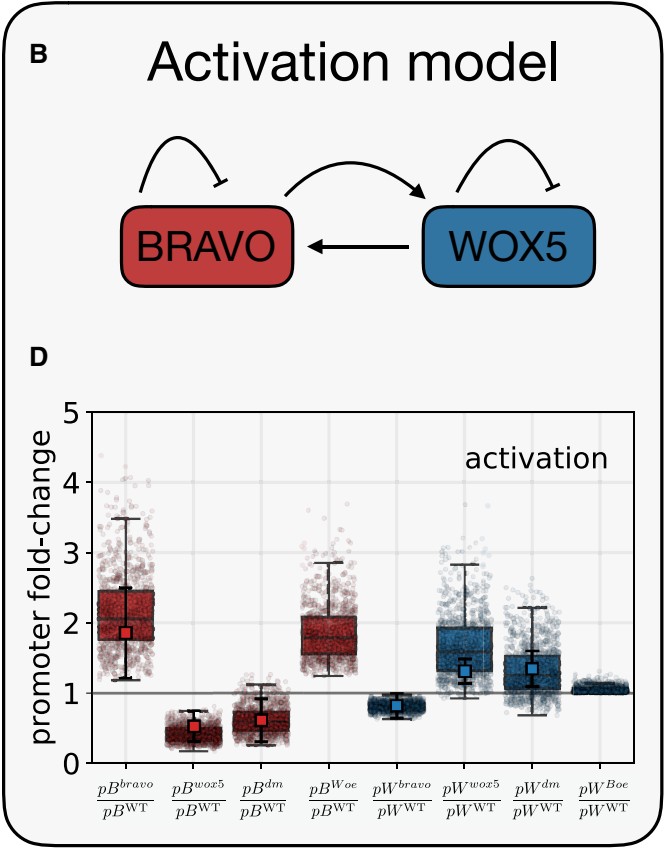

**Figure 3.   Two different models for the cross-regulations between *BRAVO* and *WOX5* in the SCN.**

A, B   Schematic representation of the effective regulations in the SCN between *BRAVO* and *WOX5* in the Alleviation (A) and Activation (B) models. (A) In the alleviation model, *BRAVO* feeds back on its own expression by reducing it and is activated by *WOX5*. *WOX5* also feeds back on its own expression by reducing it, a regulation that becomes partially impaired by *BRAVO*. (B) In the activation model, *BRAVO* is also able to activate *WOX5* which negatively feedbacks on its own expression. *BRAVO* is regulated by *WOX5* as in the alleviation model.

C, D   Parameter space exploration of (C) the alleviation and the (D) activation models. Boxplots showing the distribution of stationary promoter pBRAVO (*pB*) and pWOX5 (*pW*) fold-changes in each mutant and overexpression lines over the WT obtained for different parameter values. Superindexes denote the genotype, standing *bravo* for *bravo* mutant, *wox5* for *wox5* mutant, *dm* for the *bravo wox5* double mutant, *Woe* for WOX5 overexpression and *Boe* for BRAVO overexpression. Box width represents the interquartile range (IQR = Q3-Q1), with the horizontal line denoting the median, while whiskers extend from Q1-1.5IQR to Q3+1.5IQR. The results are obtained by solving $N = 1{,}000$ different runs of each model at the stationary state (red and blue stripplots) for the WT, mutants and overexpressor scenarios. These N runs differ in the parameter values, which are chosen at random from a uniform distribution between $P_O/2$ and $2P_O$, where $P_O$ is the default set of non-dimensional parameter values (see Appendix Table S1). For each run, the WT, mutants and overexpressor scenarios are computed all with the same parameter values. Circles denote the fold-change obtained in each run, which indicates the fold-change predicted by the model for that parameter set. These are to be compared with the experimental values of the fold-changes of the mean expressions in the mutants and overexpressor lines over the mean WT expression, which are represented by the red and blue squares overlaying the box plots, with error bars denoting ± standard deviations. The experimental values are computed from the same data as in Fig 2H and I, Appendix Figs S3 and S4). The horizontal grey line at promoter fold-change = 1 is indicated to visually separate the region of fold-change < 1 (i.e. the promoter activity is reduced in the mutant or in the overexpressor line) from the region where the fold-change > 1 (i.e. the promoter activity is increased in the mutant or in the overexpressor line).

Methods) and RNA sequencing (RNA-seq) of the root tip (Clark *et al*, 2019). Moreover, the two models indicated that the trends in the changes of expression levels between each mutant and the WT (i.e. whether the fold-change is above or below 1) are maintained when the rate of BRAVO promoter activity decreases and/or the rate of WOX5 promoter activity is increased (Appendix Fig S8C–F), mimicking the results obtained upon Brassinolide (BL, the most active BR hormone compound) treatment (Appendix Figs S4 and S5).

Exploration of the parameter space around the default parameter set (Appendix Table S1) indicated that both models can reproduce the trends of changes in *WOX5* promoter expression in the mutants

and in overexpression lines in a large parameter space (Fig 3C and D, Appendix Fig S9). Yet, the alleviation model performs better than the activation model. In larger parameter regions, this latter model can predict fold-changes that are opposite to those found in the experiments, especially in the *bravo wox5* mutant (Fig 3C and D, Appendix Fig S9). Taken together, the alleviation and the activation models reflect two, non-exclusive hypotheses for the mutual regulations between *BRAVO* and *WOX5* which can reproduce the changes of their expressions we found in the mutants and overexpressing lines. The alleviation model is more robust than the activation model at recapitulating these trends.

## BRAVO and WOX5 directly interact in a heterodimeric complex

Our results so far support that *BRAVO* and *WOX5* reinforce each other at the SCN. To further decipher how BRAVO and WOX5 interplay, we next evaluated the possible physical interaction between the BRAVO and WOX5 proteins. Using Förster resonance energy transfer measured by fluorescence lifetime microscopy (FRET-FLIM; Fig 4A–K) and yeast two-hybrid assays (Fig 4L, Appendix Fig S6A), we observed that BRAVO can interact with WOX5 (Fig 4B, G, K and L), indicating that BRAVO and WOX5 form a transcriptional complex.

We previously demonstrated that the BR-regulated BES1/TPL complex acts as a transcriptional repressor of BRAVO transcription (Vilarrasa-Blasi *et al*, 2014; Espinosa-Ruiz *et al*, 2017), in addition to BES1 directly interacting with BRAVO (Vilarrasa-Blasi *et al*, 2014). Furthermore, TPL is shown to interact with WOX5 (Pi *et al*, 2015).

Therefore, we further investigated the binding of BRAVO and WOX5 to these transcriptional regulators. We found that both BRAVO and WOX5 physically interact with BES1, and this interaction was stronger for the active BES1-D protein (Yin *et al*, 2002; Fig 4C, D, H, I and K), consistent with our previous findings that the BES1 EAR domain is necessary for BES1/BRAVO interaction (Vilarrasa-Blasi *et al*, 2014; Appendix Fig S6A). Our analysis shows that BES1 also binds to WOX5 (Fig 4H, I and K, Appendix Fig S6C), and that this interaction is stronger with BES1-D (Fig 4K). Moreover, both BRAVO and WOX5 were also observed to interact with the co-repressor TPL (Fig 4E, J, K and L, Appendix Fig S6). Collectively, these data show that BRAVO and WOX5 directly interact forming a heterodimeric complex, and that each can bind active BES1 and TPL, suggesting these proteins are able to compete for their mutual binding.

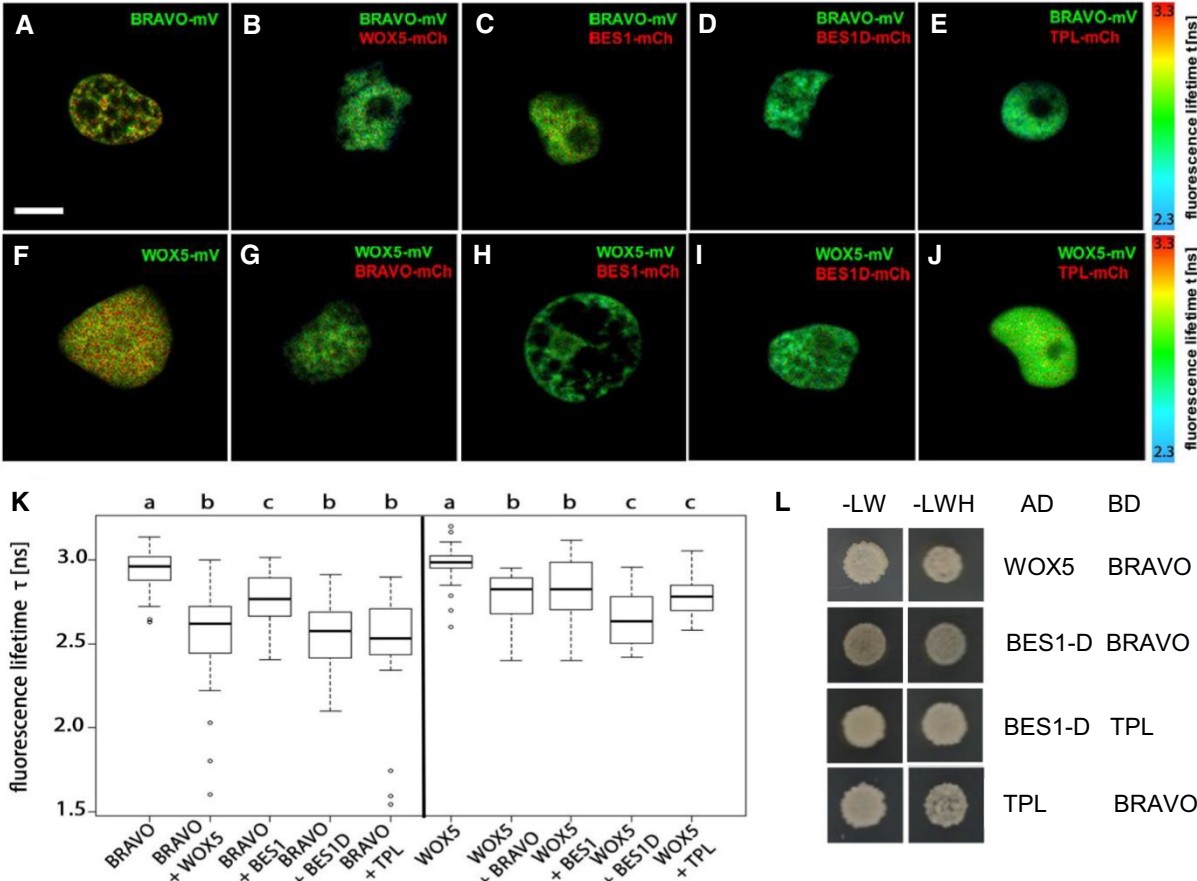

**Figure 4. BRAVO interacts with WOX5.**

A–J   Interaction of BRAVO with WOX5 (B), BES1 (C), BES1-D (D) and TPL (E); and interaction of WOX5 with BRAVO (G), BES1 (H), BES1-D (I) and TPL (J) measured by FRET-FLIM. GFP fluorescence lifetime τ [ns] was measured in transiently expressing *Nicotiana benthamiana* leaf epidermal cells. GFP fluorescence lifetime fitted pixel-wise with a mono-exponential model of BRAVO and WOX5 interactions. mV, mVenus; mCh, mCherry. Scale bar: 5 μm.

K   Fluorescence-weighted average lifetimes of BRAVO and WOX5 interactions fitted with a double-exponential model of the indicated samples are summarized in box plots. Statistical significance was tested by one-way ANOVA with a Sidakholm post hoc test. Different letters indicate statistically significant differences (*P*-value < 0.01). For each combination, two to three independent experiments were carried out, and in total, more than 20 biological replicates were measured. In the boxplot, box width represents the interquartile range (IQR = Q3-Q1), with the horizontal line denoting the median, while whiskers extend from Q1-1.5IQR to Q3+1.5IQR. White dots are the outliers.

L   Yeast two-hybrid assay showing BRAVO interacting with WOX5, BES1-D and TPL; and BES1-D interacting with TPL. In the left column, yeast cells were grown on control media, and in the right column yeast cells were grown on control media lacking Leu, Trp and His, indicating an interaction between the proteins.

                                                                                      

## The BRAVO-WOX5 complex provides a mechanism for the alleviation model

The interaction between BRAVO and WOX5 into a heterodimeric complex readily suggests a mechanism to drive the alleviation hypothesis: BRAVO, by binding to WOX5, prevents WOX5 from repressing its own expression. This assumes that the BRAVO-WOX5 complex is not able to repress *WOX5* expression. Hence, BRAVO alleviates WOX5 self-inhibition by means of a post-transcriptional sequestering mechanism. We modelled this scenario (Materials and Methods) and used as variables the amount of BRAVO protein (B), WOX5 protein (W) and the BRAVO-WOX5 protein complex (CBW) at the QC (Materials and Methods). The model assumed that BRAVO, but not the complex, inhibits its own transcription, and that WOX5, but not the complex, activates BRAVO transcription and inhibits its own transcription (Appendix Fig S10A). Notice that these regulations could take place through intermediate molecules, which are not modelled. Therefore, the model included only all those regulations that are common to the alleviation and activation models (i.e. *BRAVO* and *WOX5* self-regulations and WOX5 activates *BRAVO*) and added that BRAVO and WOX5 can form a complex with no transcriptional action on the promoters of *BRAVO* and *WOX5*. Hence, in this model, BRAVO impinges on WOX5 only through the formation of the BRAVO-WOX5 complex. Analysis of this model showed that the formation of BRAVO-WOX5 complex can account for the decreased expression of *WOX5* promoter in the *bravo* mutant and at the same time for a not very strong increase of *WOX5* under BRAVO overexpression (Appendix Fig S10A and B). Yet, while in the BRAVO overexpressing line we did not find a significant increase of *WOX5* expression (Appendix Fig S3), the model indicated that a small increase can be expected if BRAVO is set in very large amounts since it would sequester all WOX5 (Appendix Fig S10A). Together, these results support that the formation of the BRAVO-WOX5 complex can be a mechanism for BRAVO to regulate WOX5 by alleviating WOX5 self-inhibition.

We additionally included the possible competition of other factors (modelled by a new single variable S, representing e.g. TPL, BES1-D or both) for binding BRAVO and WOX5 (Appendix Fig S10C). In the model, S binds either BRAVO or WOX5, but not the BRAVO-WOX5 complex, and the heterodimer formed with BRAVO or WOX5 does not act on *BRAVO* nor *WOX5* promoter (Materials and Methods). Therefore, by binding to BRAVO or WOX5, S impedes these two transcription factors from regulating their transcriptions. The modelling results show that the mechanism of BRAVO driving alleviation of WOX5 self-inhibition holds as well in the presence of this competing factor, as long as the levels of the competing factors remain low enough to allow BRAVO and WOX5 to perform their functions (Appendix Fig S10C–E). In addition, the presence of the competing factors can reduce the effect of BRAVO overexpression on *WOX5* expression (Appendix Fig S10A and C), in agreement with the experimental data. Moreover, analysis of the model revealed that the cross-regulations between *BRAVO* and *WOX5* expressions remain less sensitive to the presence of the sequestering factor S when this competing factor can bind both factors separately than if it can bind only one of them (Appendix Fig S10C–E). This is more relevant for BRAVO-mediated regulations, since BRAVO is produced in a lesser amount than WOX5.

## BRAVO-WOX5 complex is sufficient for the control of QC divisions

The observed interactions between BRAVO and WOX5 suggest several explanations to the phenotype of QC division. For instance, the BRAVO-WOX5 heterodimer may be the molecular element that is required to repress QC divisions, or it can be either BRAVO or WOX5, since they sustain each other. To distinguish among these possibilities, we defined a regulatory function for the frequency of divided QCs and assumed that BRAVO and WOX5 can control QC division through three different contributions: one mediated by BRAVO ($T_B$), one mediated by WOX5 ($T_W$) and one mediated jointly by BRAVO and WOX5 together (hereafter named "joint contribution", $T_{BW}$). This latter contribution could take place through the BRAVO-WOX5 heterodimer. In the contribution mediated by BRAVO ($T_B$) and in that mediated by WOX5 ($T_W$), we considered that *pBRAVO* and *pWOX5* depend on each other. Specifically, we defined that the WOX5-mediated contribution to QC division in the *bravo* mutant changes by a factor of $q_W^{bravo}$ compared to the WT (Materials and Methods). Similarly, we assumed that the BRAVO-mediated contribution in the *wox5* mutant changes by a factor of $q_B^{wox5}$ compared to the WT. Small positive $q_W^{bravo}$ values ($0 < q_W^{bravo} \ll 1$) mean that BRAVO strongly upregulates the WOX5-mediated contribution to QC division. Analogously, small positive $q_B^{wox5}$ values ($0 < q_B^{wox5} \ll 1$) mean that WOX5 strongly upregulates the BRAVO-mediated contribution to QC division.

The values of $q_W^{bravo}$ and $q_B^{wox5}$ are related to the changes in the amounts of BRAVO and WOX5 proteins in the single mutants and to how these changes impinge on the regulation these proteins may have on QC division. Because the extent of these changes and hence the values of $q_W^{bravo}$ and $q_B^{wox5}$ cannot be measured, we estimated their values (see Materials and Methods for details). We set $q_W^{bravo} = 0.8$, which is similar to the fold-change of *pWOX5:GFP* expression in the *bravo* mutant compared to the WT and means that BRAVO slightly increases the contribution of WOX5 to QC division. Results were evaluated for different values of $q_W^{bravo}$.

We used the experimental data on the frequency of divided QCs in the WT, the single mutants and the double mutant (Fig 1 E), with an estimation of their confidence intervals (Materials and Methods), to extract each individual contribution (i.e. the BRAVO-mediated and the WOX5-mediated) as well as the joint BRAVO-WOX5 contribution in the WT (Materials and Methods). The results suggest that the contribution mediated by WOX5 is negligible (Fig 5A). Instead, the contribution mediated by BRAVO can be relevant if it is very strongly controlled by WOX5 (i.e. for small $q_B^{wox5}$ values) (Fig 5A). This scenario corresponds to BRAVO acting downstream of WOX5 to repress QC divisions. In this case, the model indicates that the joint contribution of BRAVO and WOX5 can also be relevant to the regulation of QC divisions in the WT, and could even facilitate QC divisions (Fig 5A). For intermediate and large $q_B^{wox5}$ values ($q_B^{wox5} > 0.4$ upwards, with $q_B^{wox5} = 0.5$ being the estimate from fold-change *BRAVO* expression in the *wox5* mutant), the model results show that in the WT the joint contribution of BRAVO-WOX5 is the only one relevant, being the contributions of BRAVO and of WOX5 very small (Fig 5A). This suggests that the joint BRAVO-WOX5 contribution is sufficient to describe the QC division phenotype independently of how strong the cross-regulations between BRAVO and WOX5 are (Fig 5B).

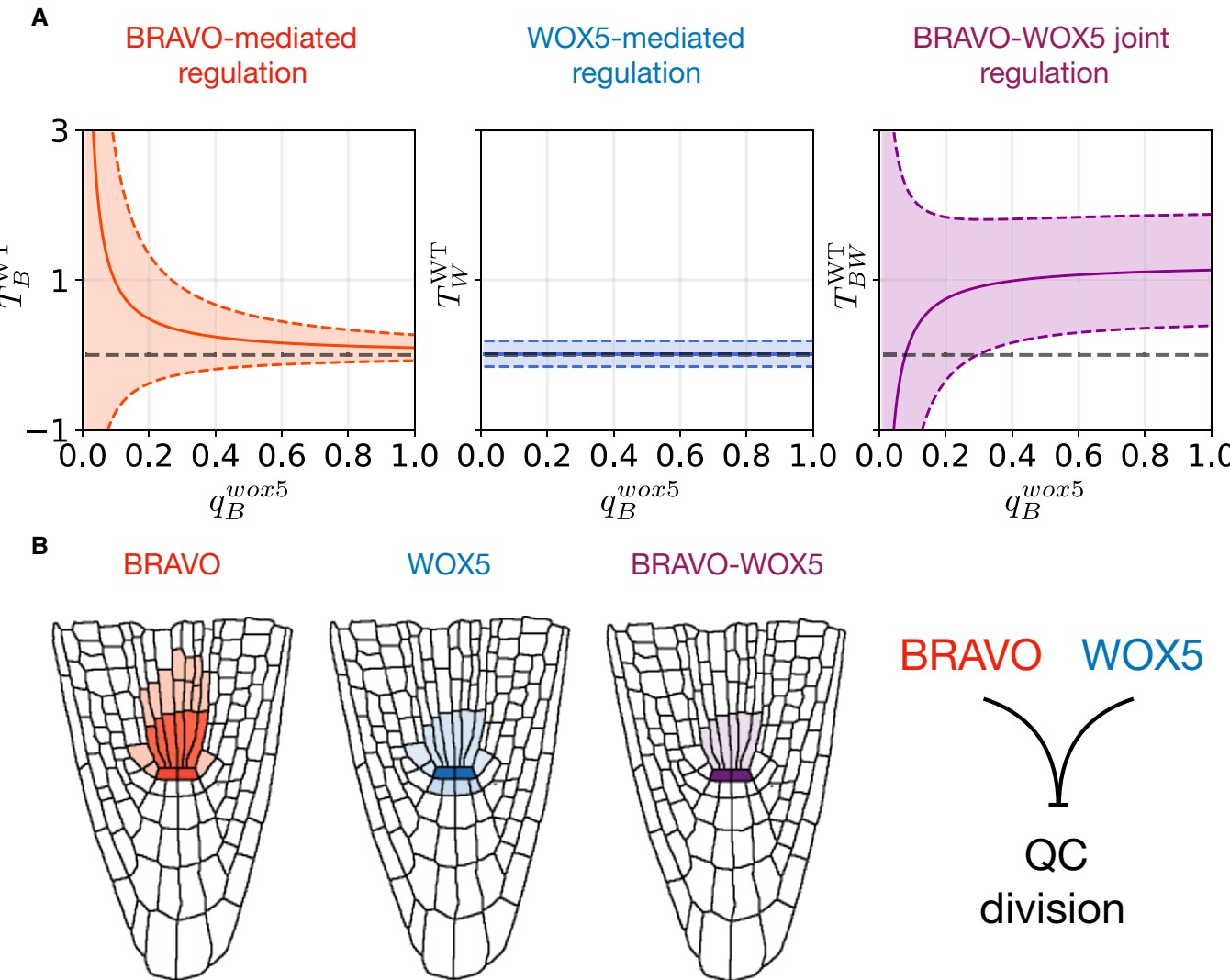

**Figure 5. BRAVO and WOX5 have a joint role in repressing QC divisions.**

A   Computational estimation of the contributions of BRAVO-mediated ($T_B^{WT}$), WOX5-mediated ($T_W^{WT}$) and BRAVO-WOX5 joint ($T_{BW}^{WT}$) regulations of QC divisions in the WT, as a function of the attenuating factor of BRAVO contribution in the *wox5* mutant, $q_B^{wox5}$. Continuous lines represent the best estimated values, while dashed lines are the enveloping confidence intervals (e.g. $T_B^{WT} \pm \delta T_B^{WT}$). The horizontal grey dashed lines mark the zero lines. For a wide range of $q_B^{wox5}$ values, the joint contribution of BRAVO and WOX5 is the only one relevant. The individual contribution of BRAVO becomes significant only for small values of $q_B^{wox5}$. In all three panels, we set $q_W^{bravo} = 0.8$. Positive contributions correspond to repression of QC divisions, while negative contributions correspond to activation of QC divisions.

B   Sketch representing the spatial distribution of BRAVO, WOX5 and their product BRAVO x WOX5, which can be interpreted as the protein complex. Their joint interaction peaks at the QC, where repression of cell division occurs.

## BRAVO and WOX5-mediated transcriptional control in the QC

Our previous results point to a joint role of BRAVO and WOX5 in the QC cells. To further understand their action, Fluorescent Activated Cell Sorting (FACS) coupled to RNA-seq was used in *pWOX5: GFP* expressing roots to isolate and sequence the GFP-marked QC cells in WT, *bravo* and *wox5* mutant backgrounds (Fig 6A; Clark *et al*, 2020). By comparing WT versus *bravo* and WT versus *wox5* RNA-seq data, we identified what we named as BRAVO- and WOX5-regulated genes, respectively.

In the QC, there were 1,472 BRAVO- and 985 WOX5-regulated genes (Fig 6B). Among them, 380 were both BRAVO- and WOX5-

regulated genes (Fig 6B and C). These data were used to evaluate if BRAVO and WOX5 have a regulatory role on downstream QC gene expression. For that, we analysed the genes that are similarly regulated by both TFs, as these may be the ones regulated by the complex. To deeper understand the behaviour of commonly deregulated genes in *bravo* and *wox5* mutants, cluster analysis was done based on their expression fold-change in both mutants (Fig 6C). When considering the genes that showed a similar expression pattern in both mutants, only a set of 53 genes was found to be upregulated (Fig 6D). This suggests that those genes might be regulated by the complex, or at least, that they are regulated by BRAVO and WOX5 in the same manner.

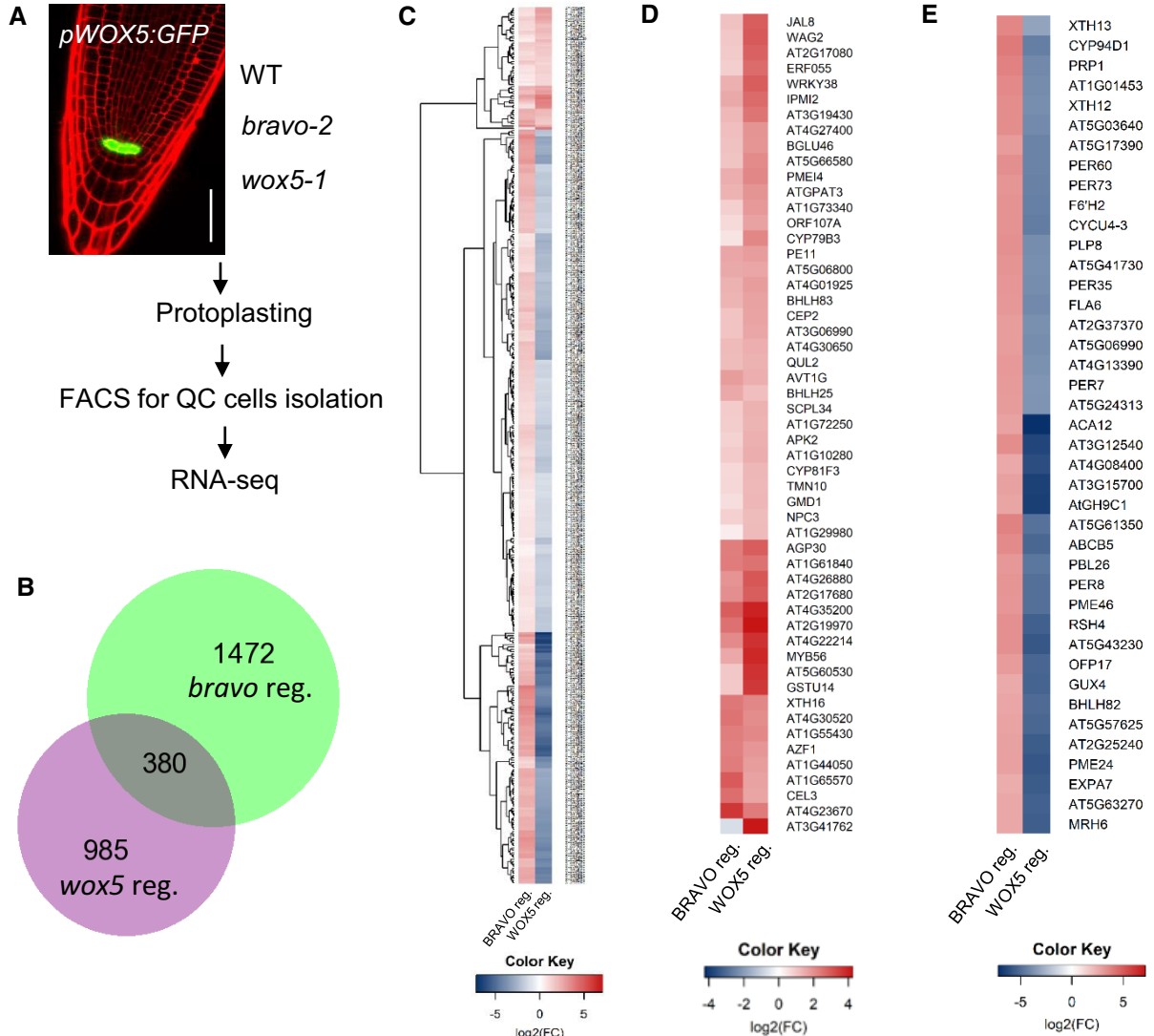

**Figure 6. Transcriptional profile of *bravo* and *wox5* QC cells.**

A  Scheme of the methodology used for the QC-specific RNA-seq. Plants expressing *pWOX5:GFP* in WT, *bravo-2* and *wox5-1* background were used for protoplasting followed by Fluorescent Activated Cell Sorting (FACS) to isolate QC cells for RNA sequencing.

B  Area-proportional Venn diagram showing the overlap between BRAVO and WOX5-regulated genes in the QC (*q* value < 0.05 and FC > 1).

C  Expression of the 380 common BRAVO and WOX5-regulated genes from (B).

D  Expression of the 53 genes from (C) showing similar expression in *bravo* and *wox5* mutants.

E  Expression of the 41 genes from (C) showing opposite expression levels in *bravo* and *wox5* mutants.

Data information: Left column in (C, D and E) heatmaps shows expression WT versus *bravo* comparisons. Right column shows expression in WT versus *wox5* comparisons. Colour bar: log2 of the fold-change.

One of the genes that appears deregulated in *bravo* and *wox5* is BRAVO (Fig 6D). In agreement with our previous results (Fig 2), WT plants have a higher BRAVO expression compared to *wox5* mutants. Regarding the rest of genes showing a similar expression pattern in *bravo* and *wox5* gene expression data, some of them have been previously characterized. Some examples are CYSTEINE ENDOPEPTIDASE 2 (CEP2), which is involved in PCD expressed in the root tip (Helm *et al*, 2008), AT4G30520 is SENESCENCE-ASSOCIATED RECEPTOR-LIKE KINASE (SARK) that regulates leaf senescence (Xu *et al*, 2011) and QUASIMODO2 LIKE 2 (QUL2) is

involved in environmental-dependent stem and vascular development (Fuentes *et al*, 2010). Among the TFs that are both BRAVO- and WOX5-regulated, there are ERF055, BHLH83 which is ROOT HAIR DEFECTIVE 6 involved in root hair initiation (Mendoza & Alvarez-Buylla, 2000), WRKY38 related to basal defence (Kim *et al*, 2008) or *ARABIDOPSIS* ZINC-FINGER PROTEIN 1 (AZF1), which acts as a transcriptional repressor involved in the inhibition of plant growth under abiotic stress conditions (Kodaira *et al*, 2011). These TFs are possible interactors of BRAVO and WOX5 participating in the same transcriptional complex. Many of the common deregulated

genes are uncharacterized, such as AT4G35200, AT2G17680, AT4G35200, AT2G19970, AT4G22214 and AT4G23670.

The number of genes showing different expression pattern in *bravo* and *wox5* mutants is higher (Fig 6C). Among them, the ones with the highest fold-change are ACA12, which is a functional $Ca^{2+}$-ATPase (Limonta *et al*, 2014), AT3G12540, AT4G08400 (EXTENSIN 7), AT3G15700 and AtGH9C1 (GLYCOSYL HYDROLASE 9C1; Fig 6 E). Most of them have not been characterized and their role in the stem cell niche remains unknown. Interestingly, several peroxidases such as PER60, PER73, PER35, PER7 and PER8 are regulated by BRAVO and WOX5 in the QC (Fig 6E). Peroxidases are proteins with no catalytic activity that are crucial to maintain redox homeostasis. In the stem cells, redox homeostasis is tightly controlled, and lower levels of ROS are found compared to the rest of cells (Wang *et al*, 2013). Whether BRAVO and WOX5 function in the stem cell niche by regulating redox homeostasis remains to be further investigated.

## Discussion

In the *Arabidopsis* primary root, the transcription factors BRAVO and WOX5 repress QC divisions and co-localize mostly at the QC (Forzani *et al*, 2014; Vilarrasa-Blasi *et al*, 2014). Our results show that BRAVO and WOX5 interact at different levels to repress QC divisions. In addition, we show that the joint action of these cell-specific transcription factors promotes overall root growth and development.

Our data indicate that BRAVO and WOX5 mutually control their expression at the SCN. While WOX5 is able to induce *BRAVO*, the impact of BRAVO on *WOX5* is less severe. This is also supported by the RNA-seq data where BRAVO expression is downregulated in *wox5* mutant, but no significant change in WOX5 expression is found in *bravo* mutant (Fig 6). This suggests that the effect of WOX5 on *BRAVO* and thereby on BRAVO-mediated regulation can be more relevant than the effect BRAVO has upon *WOX5* and WOX5-mediated action. This is consistent with the known SCN phenotypes of *bravo* and *wox5* mutants (Sarkar *et al*, 2007; Bennett *et al*, 2014; Forzani *et al*, 2014; Vilarrasa-Blasi *et al*, 2014; Pi *et al*, 2015), where *wox5* exhibits, besides a similar increased QC division phenotype as *bravo*, an overall distorted and disorganized SCN morphology and CSC premature differentiation that is absent in the *bravo* mutant.

Another important molecular link between BRAVO and WOX5 revealed by our data is their physical protein-protein interaction. The two transcription factors mostly co-localize at the QC, suggesting that they act as co-partners of a single complex only in the QC, where they converge. Through modelling, we show that the existence of this protein complex provides a natural way to explain how BRAVO regulates *WOX5* expression at the QC. According to this mechanism, BRAVO partially sequesters WOX5 into a complex and impedes *WOX5* self-repression. This alleviation mechanism does not exclude the activation of *WOX5* by BRAVO to take place as well. Moreover, our data revealed that additional factors, such as TPL and BES1, can bind to BRAVO and WOX5 and compete for their binding. According to our modelling results, the fact that these competing factors can bind separately either BRAVO or WOX5 makes the cross-regulation between *BRAVO* and *WOX5* expressions

less sensitive to the amount of competing factors. This suggests that promiscuity of protein-interacting partners can be a mechanism to enable modular regulations of gene expression.

All the interactions set in the models between *BRAVO* and *WOX5* expressions at the SCN are effective in the sense that they are the result of multiple, direct and indirect, regulatory mechanisms. For instance, *WOX5* self-repression can involve a negative feedback where WOX5 activates a repressor or represses an activator, among other possibilities. In this context, control of auxin-ARF and auxin-IAA (Tian *et al*, 2014b) as well as the PLETHORA genes (preprint: Burkart *et al*, 2019) were all shown to involve negative feedback loops with *WOX5*. WOX5 induction of *BRAVO* expression could be as well through a downstream target of WOX5. Moreover, BRAVO is found to ultimately down-regulate its own expression, although this probably occurs through other intermediate molecules, as BRAVO has been previously shown to activate itself by directly binding its own promoter (Vilarrasa-Blasi *et al*, 2014).

Our analysis supports that BRAVO-WOX5 joint regulation is sufficient to account for the QC division phenotype. This joint regulation could take place by several means, yet the finding that BRAVO and WOX5 can interact together suggests that this complex can be mediating it. At a mechanistic level, the BRAVO-WOX5 protein complex may bind CYCLIN-D3:3, as shown to occur for WOX5 (Forzani *et al*, 2014). Although the joint action is sufficient, other scenarios cannot be discarded. For example, BRAVO could individually repress QC divisions, while being strongly upregulated by WOX5. This scenario proposes that BRAVO acts downstream of WOX5 to repress QC divisions. The data support that in this case, the joint action of BRAVO and WOX5 could be relevant and could even induce QC divisions. The case where BRAVO is the only factor repressing QC divisions and the joint BRAVO-WOX5 contribution is negligible cannot be completely discarded either.

Interestingly, we also found that BRAVO and WOX5 promote root growth and lateral root development. In lateral root development, the formation of the organizing centre and the stem cell niche occurs after lateral root initiation (Banda *et al*, 2019). A high number of genes are commonly expressed at the SCN of primary and lateral roots, such as PLT, SHR, SCR or TCP (Goh *et al*, 2016; Shimotohno *et al*, 2018). Loss-of function of these genes leads to an increased number of aberrant lateral roots and reduced levels of *WOX5* (Shimotohno *et al*, 2018), and thus, it is possible that BRAVO and WOX5 not only control stem cell niche maintenance in the primary root, but also in the lateral roots. Thus, the consistent and overlapping role of BRAVO and WOX5 at promoting lateral root development also points to a relevant role of the BRAVO-WOX5 complex for this function.

Transcriptomic analysis of QC cells in *bravo* and *wox5* mutants reveals a common set of genes regulated by both TFs. In addition, the observed predominant role of BRAVO acting as an activator and WOX5 as a repressor of transcription points to a downstream compensatory mechanism. Along those lines, we found several components involved in redox homeostasis being regulated by BRAVO and WOX5 in a different manner. Remarkably, gene function related to cell division was not found, suggesting that their role in this process is achieved through the regulation of yet uncharacterized genes. Our RNA-seq data provide the transcriptional landscape downstream BRAVO and WOX5 with cell-type resolution, and further characterization of the identified processes and genes can

shed light on the molecular mechanisms downstream those TF in the SCN homeostasis.

To conclude, understanding signalling networks operating in stem cell development is becoming essential to decipher plant growth and adaptation to the environment. Systems biology approaches provide a closer picture to reality unveiling how complex and dynamic networks of cell-specific transcription factors act to preserve stem cell function in plants. Here, untapping the action of two main regulators of quiescent cell division, BRAVO and WOX5, not only indicates that these factors operate as a transcriptional complex in preserving stem cell function, but also unveils their joint roles in primary and lateral root development.

# Materials and Methods

### Plant material and root measurement

All WT, mutants and transgenic lines are in the *Arabidopsis* ecotype Columbia (Col-0) background (Appendix Table S2). The double mutant *bravo wox5* was generated by crossing the *bravo* and *wox5* single mutants. The double mutant homozygous lines were selected by genotyping. The primers used for *bravo* and *wox5* genotyping are listed in Appendix Table S3. Seeds were surface sterilized and stratified at 4°C for 48 h before being plated onto 0.5× Murashige and Skoog (MS) salt mixture without sucrose and 0.8% plant agar, in the absence or presence of Brassinolide (Wako, Osaka, Japan). β-estradiol (30 μM) from Sigma diluted in DMSO was used to induce BRAVO expression for 6 days. Dexamethasone (1 μM) from Sigma diluted in EtOH was used to induce WOX5 expression for 6 days. For RT–qPCR experiments, β-estradiol and dexamethasone treatments were applied for 24 h. Plates were incubated vertically at 22°C and 70% humidity in a 16 h light/8 h dark cycle. Primary root length was measured from plates images, using ImageJ (https://imagej.nih.gov/ij/) and MyROOT (Betegon-Putze *et al*, 2019) software. The lateral root density was calculated by dividing the total number of emerged lateral roots of individual seedlings by the mean of the root length of those seedlings.

### Confocal microscopy and quantification of fluorescence signal

Confocal images were taken with a FV 1000 Olympus confocal microscope after Propidium iodide (PI, 10 μg/ml) staining. PI and GFP were detected with a bandpass 570–670 nm filter and 500–545 nm filter, respectively. Images were taken in the middle plane of 6-day-old roots. The fluorescence intensity was quantified with ImageJ using the Integrated Density value obtained from individual plants. The quantified area was selected with a ROI that contained the SCN (Appendix Fig S7). The laser settings for *pBRAVO*:GFP and *pWOX5*:GFP are different, as WOX5 has a stronger expression than BRAVO. The laser intensity applied was higher for *pBRAVO:GFP* than for *pWOX5:GFP* lines. The analysis of *pBRAVO*:GFP in *bravo wox5* double mutant background was done with different confocal settings. The analysis of QC cell division and CSC differentiation was carried out by imaging fixed roots through a modified pseudo-Schiff (mPS-PI) staining method (Truernit *et al,* 2008). Images were processed with the Olympus FV (Olympus, Tokyo, Japan) and ImageJ software.

### RT–qPCR assay

RNA was extracted from root tip tissue with the Maxwell® RSC Plant RNA Kit (Promega) using the Maxwell® RSC instrument (Promega) according to the manufacturer's recommendations, and concentrations were checked using NanoDrop 1000 Spectrophotometer (Thermo Fisher Scientific). cDNA was obtained from RNA samples by using the NZY First-Strand cDNA Synthesis Kit (NZYtech) according to the manufacturer's recommendations. RT–qPCR amplifications were performed from 10 ng of cDNA using SYBR Green I master mix (Roche) in 96-well plates according to the manufacturer's recommendations. The RT–qPCR was performed on a LightCycler 480 System (Roche). *ACTIN2* (AT3G18780) was used as housekeeping gene for relativizing expression. Primers used are described in Appendix Table S3.

### Yeast two-hybrid assay

Yeast two-hybrid assays were performed by the Matchmarker GAL4-based two-hybrid System (Clontech). Constructs were co-transformed into the yeast strain AH109 by the lithium acetate method (Gietz & Woods, 2002). The presence of the transgenes was confirmed by growth on SD-LW plates, and protein interaction was assessed by selection on SD-LWH plates. Interactions were observed after 4 days of incubation at 30°C.

### Transient expression in *Nicotiana benthamiana* for FLIM measurements

Preparation of transiently expressing *Nicotiana benthamiana* leaves and induction of fusion proteins tagged with either mVenus or mCherry by application of ß-estradiol was carried out as described in Bleckmann *et al* (2010).

### Acquisition of FLIM data

FLIM data acquisition was carried out using a confocal laser scanning microscope (LSM780 inverted microscope, Zeiss) equipped additionally with a time-correlated single-photon counting device with picosecond time resolution (Hydra Harp 400, PicoQuant). mVenus was excited at 485 nm with a pulsed (32 MHz) diode laser at 1.2 μW at the objective (40× water immersion, C-Apochromat, NA 1.2, Zeiss). The emitted light was collected through the same objective and detected by SPAD detectors (PicoQuant) using a narrow range bandpass filter (534/35, AHF). Images were taken at 12.5 μs pixel time and a resolution of 138 nm/pixel in a $256 \times 256$ pixel image. A series of 40 frames was merged into one image and analysed using the Symphotime software package (PicoQuant).

### Analyses and presentation of FLIM data

FRET between fluorophores with different spectra like the here used GFP and mCherry is strongly distance dependent on a nanometer length scale and therefore is an indicator for the interaction of the fused proteins (Borst & Visser, 2010). The fluorescent lifetime of the collected photons in each merged image was analysed using the Symphotime software (PicoQuant). For this, a ROI covering the whole nucleus was created to reduce background fluorescence. All photons in this ROI were used to build a histogram of the fluorescence decay.

A double-exponential fit model was used to approximate the intensity-weighted average fluorescence lifetime τ[ns] of all photons of the ROI. The instrument response function was measured with KI-quenched erythrosine and used for reconvolution in the fitting process (Weidtkamp-Peters & Stahl, 2017). The data from replicate measurements were summarized in box plots created in R software (https://www.r-project.org/). Statistical significance was tested by one-way ANOVA with a Sidakholm post hoc test. Different letters indicate statistically significant differences ($P < 0.01$).

For the creation of FLIM images, photons from individual pixels of a merged image were analysed for fluorescent lifetime using the Symphotime software (PicoQuant). A mono-exponential fit model was used, as the photon number in each pixel was too low for a double-exponential model (Stahl et al, 2013). The individual pixels are colour-coded according to their fluorescence lifetime.

## Bimolecular fluorescence complementation assay (BiFC)

The BRAVO and WOX5 coding sequences were inserted by LR-reaction (Invitrogen) into pBiFC binary vectors containing the N- and C-terminal YFP fragments (YFPN43 and YFPC43). Plasmids were transformed into the Agrobacterium tumefaciens GV3101 strain, and appropriate combinations were infiltrated into Nicotiana benthamiana leaves (Occhialini et al, 2016). The p19 protein was used to suppress gene silencing. Infiltrated leaves were imaged 2 days after infiltration using an Olympus FV1000 laser scanning confocal microscope.

## QC-specific transcriptomics

The protocol for the isolation of QC cells through FACS was described in Clark et al (2018).

For WT and bravo-2 RNA-seq data, quality control was done with FastQC (available at https://www.bioinformatics.babraham.ac.uk/projects/fastqc/). Adaptor sequences were removed using cutadapt. The mapping of the reads to the genome was done with STAR (Dobin et al, 2013) and the quantification of the reads with RSEM (Li & Dewey, 2011). Araport11 were used as transcripts of reference and TAIR10 as reference genome. In R studio, genes with 0 counts in all samples were removed. NOISeq package was used for the bias detection, correction and normalization (Tarazona et al, 2015). ARSyN was used for batch effect correction as the sorting was detected as a confounded factor (Nueda et al, 2012). Genes with low number of counts (cpm < 4) were filtered. NOISeqBIO was used for analysis of differential expression between the different conditions (Tarazona et al, 2015). Regulated genes were selected with q value < 0.05 and fold-change > 1.

The transcriptional profile of wox5-1 mutants was described in Clark et al (2020).

## Mathematical models of the effective regulations between BRAVO and WOX5 expressions

We considered two mathematical models for the effective regulations that BRAVO and WOX5 expressions perform on each other and on themselves in the SCN. The models are named Alleviation and Activation models. Both models are formulated similarly and only differ on how WOX5 expression is regulated by BRAVO. In both models, B and W are the variables denoting the total BRAVO and WOX5 expressions, respectively, in the whole SCN. Each of these expressions are considered to be the product of the corresponding promoter activity according to the following wild-type dynamics:

$$\frac{dB}{dt} = P_B(B, W) - d_B B \qquad (1)$$

$$\frac{dW}{dt} = P_W(B, W) - d_W W \qquad (2)$$

where $P_B$ (B,W) and $P_W$ (B,W) are the BRAVO and WOX5 promoter activities (production terms), respectively, and $d_B B$ and $d_W W$ are the decay terms (assumed linear for simplicity, with decay rates $d_B$ and $d_W$). To account for the regulation of the expression, each promoter activity depends on BRAVO and WOX5 expressions. To compare with empirical data, we only considered the stationary state of the above dynamics (i.e. when time derivatives are equal to zero, $\frac{dB}{dt} = 0$, $\frac{dW}{dt} = 0$). In the stationary state, BRAVO expression is proportional to BRAVO promoter activity ($B = P_B(B,W)/d_B$) and WOX5 expression is proportional to WOX5 promoter activity ($W = P_W(B, W)/d_W$). Therefore, we used the promoter activity in the stationary state as the computational model read-out to be compared with the empirical data on pBRAVO:GFP and pWOX5:GFP.

Promoter activity terms $P_B$ (B,W) and $P_W$ (B,W) correspond to functions that describe the effective regulations that each expression ultimately performs on each promoter activity (see Fig 3A and B for cartoons of these regulations for each model). These effective regulations are expected to involve several intermediate steps, including translational and post-translational processes, and additional molecules. These are not explicitly modelled but are all together absorbed in the functionalities of $P_B$ (B,W) and $P_W$ (B,W). We expect these functions to be nonlinear, and we used continuous Hill-like functions exhibiting saturation with exponents larger than 1 (see parameter values in Appendix Table S1). For both models, we consider the following functional form for the BRAVO promoter:

$$P_B(B, W) = \alpha \left( \frac{1 + \varepsilon_B (K_B B)^2}{1 + (K_B B)^2} \right) \left( \frac{1 + \varepsilon_W (K_W W)^2}{1 + (K_W W)^2} \right) \qquad (3)$$

This BRAVO promoter activity $P_B$ (B,W) has (i) a basal production rate α, independent of BRAVO and WOX5 expressions since our GFP data show that BRAVO promoter has activity in the double mutant bravo wox5 (Appendix Fig S2), (ii) a term that sets the activation of BRAVO expression by WOX5, with WOX5 expression threshold value $1/K_W$ and activation strength $\varepsilon_W > 1$. According to this term, the production of BRAVO increases to $\alpha\varepsilon_W > \alpha$ if WOX5 expression is very high ($W \gg 1/K_W$) and there is no BRAVO. (iii) A term that accounts for the inhibition of BRAVO expression by itself, with threshold value $1/K_B$ and inhibition strength $\varepsilon_B < 1$. According to this term, the production of BRAVO decreases to $\alpha\varepsilon_B < \alpha$ when BRAVO is very high ($B \gg 1/K_B$) and there is no WOX5.

For the Alleviation model, we consider the following functional form for the WOX5 promoter:

$$P_W(B, W) = \gamma \frac{W_0^2}{W_0^2 + W^2 \left( \frac{B_0^2}{B^2 + B_0^2} B_1 + 1 \right)^2} \qquad (4)$$

This WOX5 promoter activity $P_W$ has (i) a basal production in the absence of *BRAVO* and *WOX5* expressions of value $\gamma$; (ii) *WOX5* expression ultimately represses its own production. $W_0$ sets the characteristic *WOX5* threshold expression of this repression. (iii) Part of this self-repression is dependent on *BRAVO*, which reduces the strength of *WOX5* self-repression. The parameter $B_1$ sets the value of the maximum alleviation BRAVO can make on WOX5 self-repression. The parameter $B_0$ sets the *BRAVO* threshold expression for the alleviation.

For the Activation model, the promoter of WOX5 is:

$$P_W(B, W) = \gamma \frac{W_0^2}{W_0^2 + W^2} \left( \frac{B^2}{B^2 + B_0^2} B_1 + 1 \right) \tag{5}$$

This WOX5 promoter has a basal rate $\gamma$ and the type of inhibition mediated by WOX5 as the Alleviation model (equation (4), being $W_0^2$ the threshold of inhibition as in equation (4)). In addition, it includes that BRAVO activates the WOX5 promoter in a WOX5-independent manner, with a maximum strength $B_1$. This activation has a threshold $B_0$.

For clarity, here we presented the full mathematical models with all their corresponding parameters, but all the simulations were made with their non-dimensional counterparts (see Expanded View text), in order to avoid redundancies in the effect that changes in the parameter values when all parameters are changed simultaneously.

## Mathematical model that considers the formation of the BRAVO-WOX5 complex (complex formation model)

In this model, variables denote proteins. $B, W$ and $S$ denote the concentration of BRAVO protein, WOX5 protein and of an additional $S$ protein, respectively, that are not bound to each other. $S$ protein can be interpreted as TOPLESS, BES1 or other possible partners of BRAVO and/or WOX5) and is able to bind to BRAVO and to WOX5 separately. BRAVO and WOX5 are also considered to be able to bind each other. By denoting the concentrations of BRAVO-WOX5, BRAVO-S and WOX5-S heterodimers as $C_{BW}$, $C_{BS}$ and $C_{WS}$, respectively, we write the dynamics of each molecule as:

$$\frac{dB}{dt} = P_B(B, W) - \lambda_{BW}BW + \mu_{BW}C_{BW} - \lambda_{BS}BS + \mu_{BS}C_{BS} - d_B B \tag{6}$$

$$\frac{dW}{dt} = P_W(B, W) - \lambda_{BW}BW + \mu_{BW}C_{BW} - \lambda_{WS}WS + \mu_{WS}C_{WS} - d_W W \tag{7}$$

$$\frac{dS}{dt} = \beta - \lambda_{BS}BS + \mu_{BS}C_{BS} - \lambda_{WS}WS + \mu_{WS}C_{WS} - d_S S \tag{8}$$

$$\frac{dC_{BW}}{dt} = \lambda_{BW}BW - \mu_{BW}C_{BW} - d_{BW}C_{BW} \tag{9}$$

$$\frac{dC_{BS}}{dt} = \lambda_{BS}BS - \mu_{BS}C_{BS} - d_{BS}C_{BS} \tag{10}$$

$$\frac{dC_{WS}}{dt} = \lambda_{WS}WS - \mu_{WS}C_{WS} - d_{WS}C_{WS} \tag{11}$$

with $P_B(B,W)$ given by Eq. (3) and

$$P_W(B, W) = \gamma \frac{W_0^2}{W_0^2 + W^2} \tag{12}$$

In this model, the first terms in the right-hand-side of Equations (6-8) set the production (translation) of BRAVO, WOX5 and S proteins, respectively ($P_B(B,W)$, $P_W(B,W)$ and $\beta$). The above equations use the quasi-steady-state approximation for the mRNAs, which are assumed to be transcribed and linearly degraded. Therefore, the production (translation) terms are each proportional to the corresponding promoter activity. The BRAVO production term has the same features as the promoter function in the Alleviation and Activation models (i.e. activation by WOX5 and inhibition by BRAVO), while the production of WOX5 protein only considers repression by WOX5 (i.e. the common feature of the promoters in the Alleviation and Activation models). For simplicity, protein S is assumed to be translated at a constant rate $\beta$, independent of BRAVO and WOX5. The model assumes that the binding between proteins happens only pairwise (i.e. with no higher-order complexes), reactions of binding and unbinding are reversible, protein complexes (heterodimers) become degraded, and these complexes do not regulate the translation/transcription of any of the modelled proteins. In the equations, protein complexes are formed with binding rate $\lambda_{XY}$ where $X$ and $Y$ represent each possible interacting protein ($B$, $W$, $S$), unbind with unbinding rate $\mu_{XY}$ and are degraded with rate $d_{XY}$. The model results are only evaluated at the stationary state (i.e. Equations (6-11) are set to zero, see Expanded View text). All simulations were done for the dimensional model. Default parameter values are indicated in Appendix Table S1.

This model can be interpreted as a mechanistic realization of the alleviation model, as in the *bravo* mutant the amount of free WOX5 can increase due to the absence of one of its sequestering factors, thereby increasing the WOX5 self-repression.

## Modelling of the mutants and overexpressing lines

To model the mutants, we used the same equations and parameter values as for the WT with the only changes being: in the mutant background (*bravo*, *wox5* or *bravo wox5*) the expression of the mutated gene is null at all times ($B = 0$ in the *bravo* mutant, $W = 0$ in the *wox5* mutant, and both $B = W = 0$ in the *bravo wox5* mutant), despite its promoter activity is nonzero. The promoter activities are computed according to the promoter functions $P_B$ and $P_W$ as defined for the WT but evaluated at $B = 0$ in the *bravo* mutant, $W = 0$ in the *wox5* mutant, and at $B = W = 0$ in the *bravo wox5* mutant. No additional changes (e.g. no changes in parameter values) were considered to occur in the mutants. The model equations for all the mutants are detailed in the Expanded View text. Herein, we exemplify only the *bravo* mutant for the Alleviation model (where the superscript *bravo* is used to denote this mutant):

$$B^{bravo} = 0, \ P_B(0, W^{bravo}) = \alpha \frac{1 + \varepsilon_W (K_W W^{bravo})^2}{1 + (K_W W^{bravo})^2}$$

$$\frac{dW^{bravo}}{dt} = P_W(0, W^{bravo}) - d_W W^{bravo},$$

$$P_W(0, W^{bravo}) = \gamma \frac{W_0^2}{W_0^2 + (W^{bravo})^2(B_1^2 + 1)^2}$$

Overexpression was modelled through an additive constant increase in the production rate of each expression such that the overall production of *BRAVO* changed to $\alpha A_0 + P_B(B,W)$ for

overexpression of BRAVO and the overall production of *WOX5* changed to $\gamma G_0 + P_W(B,W)$ for overexpression of WOX5. Since the basal production due to the endogenous promoter is $\alpha$ and $\gamma$, respectively, parameters $A_0$ and $G_0$ set the extent of the overexpression relative to the basal production. The model outcomes, such as the expression of the regulated (not overexpressed) factor, do not increase indefinitely with this extent but show a saturating response. We chose parameter values for $A_0$ and $G_0$ sufficiently high such that the saturated response is reached.

To compare with empirical data on GFP expression in the mutants and overexpressing lines, we only considered the stationary state of all models, in the WT, mutant and overexpression scenarios (see detail in Expanded View text).

### Comparison of model outputs with empirical data on GFP expression

The values of the promoter activities (i.e. production terms), $P_B$ and $P_W$, at the stationary state (i.e. when time derivatives are equal to zero) were those used for comparison with the GFP data measured in the whole SCN. We used the notation $pB$ and $pW$ to refer to stationary *BRAVO* and *WOX5*, respectively, promoter activities (or production terms, $pB \equiv P_B(t \to \infty)$ and $pW \equiv P_W(t \to \infty)$). The super-indexes *WT*, *bravo*, *wox5* and *dm* were used to refer to the promoter in the stationary state for the WT, the *bravo* mutant, the *wox5* mutant and the double mutant, respectively (Stationary solutions section in Expanded View text). Since GFP scale is arbitrary with respect to promoter activity, we used the ratios that set the fold-change between mutant and the WT as the relevant measure to be compared between model outputs and empirical data. For the empirical data, we used the mean GFP measured values and computed the ratio of the mean GFP expression in the mutant over the mean GFP expression data in the WT, for each mutant. Error bars of fold-changes were computed using error propagation where the standard deviation of each GFP expression was set as the error. For each model, we computed the ratios of the stationary production in each mutant over its stationary production value in the WT:

$$\frac{pB^{bravo}}{pB^{WT}}, \frac{pB^{wox5}}{pB^{WT}}, \frac{pB^{dm}}{pB^{WT}}, \frac{pW^{bravo}}{pW^{WT}}, \frac{pW^{wox5}}{pW^{WT}}, \frac{pW^{dm}}{pW^{WT}}$$

Parameter values for the Activation and Alleviation models (Appendix Table S1) were chosen such that the values of these ratios obtained from the model fit the ratios computed from the mean GFP expression values (Appendix Fig S8E and F). Since the GFP data are a broad distribution, there is a broad range of parameters in which the model fits the experiments within the range of experimental deviations. In addition, the models reproduce for a wide range of parameter values whether these ratios are > 1 (i.e. in the mutant, the promoter activity increases) or < 1 (i.e. in the mutant, the promoter activity decreases).

Additionally, the outputs of the models were numerically computed for different values of $\alpha$ and $\gamma$ (all the remaining parameter values being unchanged), to model different conditions of the growth medium (Appendix Fig S8C–F). Specifically, we set $\alpha$ and $\gamma$ as functions of an auxiliary control parameter $x$ that indicates the medium condition ($x = 1$ corresponds to CTL conditions, whereas higher $x$ values correspond to a medium with BL). We used

$\alpha = 0.3/x$ and $\gamma = 250x/(x + 9)$, such that for $x = 1$ $\alpha$ and $\gamma$ take the values of the WT in CTL conditions (for $x = 1$, $\alpha$ and $\gamma$ take the values in Appendix Table S1). Roughly, $x$ controls the disparity between the basal production of *BRAVO* and *WOX5*. This allows us to interpret high values of $x$ as the effect of Brassinolide.

The stationary values of the production terms, $pB$ and $pW$, were also computed when overexpression of either BRAVO (denoted by superindex *Boe*) or WOX5 (denoted by superindex *Woe*) was applied and compared to the stationary values in the WT scenario:

$$\frac{pB^{Woe}}{pB^{WT}}, \frac{pW^{Boe}}{pW^{WT}}$$

### Numerical methods to obtain model outputs

In the stationary state (i.e. when time derivatives are equal to zero), the alleviation and activation models for the WT and overexpression lines reduce to a system of two coupled algebraic equations (see Stationary solutions in Expanded View text). For each mutant, these two models reduce each to a single algebraic equation (see Stationary solutions in Expanded View text). The complex formation model in the stationary state reduces to three coupled algebraic equations in the case of the WT and overexpressing lines, and to two coupled algebraic equations in the case of mutants (see Stationary solutions in Expanded View text). To find the stationary stable solutions, we solved these algebraic equations numerically with custom-made software and using the fsolve routine embedded in Python (Python Software Foundation, https://www.python.org/), which uses a modification of Powell's hybrid method for finding zeros of a system of nonlinear equations.

To evaluate the robustness of each model to provide the average trends of fold-changes in gene expression, we computed these fold-changes across the parameter space. To that end, we defined N different sets of parameter values. For each set, the stationary solutions of all genotypes (WT, all mutants and all overexpression lines) were computed. This enabled to obtain the fold-changes of expression predicted by each model for that set of parameter values. The predictions for each parameter set are to be compared with the mean fold-changes found in experiments. By comparing the model predictions for the N sets of parameter values (i.e. across the parameter space) with the mean values obtained in the experiments, we gain insight on whether the model is able to reproduce the mean experimental data in a large region of the parameter space or whether it is not. The N sets of parameter values were randomly chosen from uniform distributions of the parameter values around the default set of parameter values detailed in Appendix Table S1. The range of these uniform distributions is detailed in Figure captions. For the Activation and Alleviation models, the parameter sets were defined for the non-dimensional parameters, and each set differed in the values of all non-dimensional parameters simultaneously (Fig 3C and D, Appendix Figs S8E and F, and S9C and D). To evaluate the sensitivity to each parameter, sets which differ only in the value of a single non-dimensional parameter were also studied (Appendix Fig S9A and B). For the complex formation model, the N parameter sets were defined for the dimensional parameter values. Each set differed in all parameters except for all the degradation rates and the unbinding coefficients that were the same in all sets (Appendix Fig S10). These latter parameters were not allowed to

vary because they drive redundant outcomes to the stationary solutions (see Stationary solutions in Expanded View text).

## Estimation of the error in the QC division data

We denote by $a,b,c$ and $d$ the values that we obtain empirically for the percentage of roots that exhibit a divided QC in the WT, the *bravo* mutant, the *wox5* mutant and the double *bravo wox5* mutant, respectively ($a = 0.3939$, $b = 0.8732$, $c = 0.8070$, $d = 0.8846$). We can estimate the error in each of these measures, by assuming our measurement for each genotype corresponds to $N$ independent equivalent roots where we observe whether the QC exhibits any division or not (i.e. we have $N$ independent Bernoulli experiments). By assuming that the probability of observing a QC with at least one cell divided is $p$ ($p = a,b,c,d$ for each of the genotypes under study), we can estimate the error. Specifically, we assumed $p = N_k/N$, where $N_k$ is the number of roots, from the total $N$ of the specific genotype, that have a divided QC and set the error as the standard deviation of $p = \frac{N_k}{N}$: $\delta p \equiv std\left(p = \frac{N_k}{N}\right) = \sqrt{\frac{p(1-p)}{N}}$. For each genotype, we took a conservative view and used $N = 15$ for computing the errors, to avoid their underestimation.

## A model to compute the contribution of BRAVO and WOX5 to regulate QC division

We aim at evaluating the contribution of BRAVO and WOX5 on regulating QC divisions. To this end, we propose the following function:

$$F = \frac{F_0}{1 + T_B + T_W + T_{BW}}$$

which indicates the frequency at which we found a QC with at least one QC cell that is divided in the plane of observation, for roots of the same genotype. $T_B$, $T_W$ and $T_{BW}$ are the contributions mediated by BRAVO, by WOX5 and jointly by both BRAVO and WOX5, on the regulation of QC division. Notice each contribution corresponds to repression of QC divisions when it takes positive values. In contrast, it corresponds to induction of QC divisions for negative values. This function can be applied to the WT, to each single mutant and to the double mutant, such that in the *wox5* mutant we have $T_W^{wox5} = 0$ and $T_{WB}^{wox5} = 0$, while in the *bravo* mutant we have $T_B^{bravo} = 0$ and $T_{WB}^{bravo} = 0$. Thus, this function takes the following expressions in the WT and in the mutants:

$$F^{WT} = \frac{F_0}{1 + T_B^{WT} + T_W^{WT} + T_{BW}^{WT}}$$

$$F^{bravo} = \frac{F_0}{1 + T_W^{bravo}} = \frac{F_0}{1 + T_W^{WT} q_W^{bravo}}$$

$$F^{wox5} = \frac{F_0}{1 + T_B^{wox5}} = \frac{F_0}{1 + T_B^{WT} q_B^{wox5}}$$

$$F^{dm} = F_0$$

where superindexes WT, *bravo*, *wox5* and *dm* account for WT, *bravo* mutant, *wox5* mutant and *bravo wox5* double mutant, respectively. In these expressions, we also defined $T_W^{bravo} \equiv T_W^{WT} q_W^{bravo}$ and $T_B^{wox5} \equiv T_B^{WT} q_B^{wox5}$. Hence, parameter $q_B^{wox5}$ measures the change in the

strength of the BRAVO-mediated contribution to QC division in the *wox5* mutant ($T_B^{wox5}$) compared to its strength in the WT ($T_B^{WT}$). Analogously, $q_W^{bravo}$ parameter measures the change in the strength of the repression that WOX5 does on QC division in the *bravo* mutant ($T_W^{bravo}$) compared to the strength it does on the WT ($T_W^{WT}$). Notice that we assumed no additional changes happen in the $F$ function in these mutants.

We can extract the values of $T_B^{WT}$, $T_W^{WT}$ and $T_{BW}^{WT}$ by equating the above expressions to the empirical data ($F^{WT} = a$, $F^{bravo} = b$, $F^{wox5} = c$, $F^{dm} = d$) and once values to $q_W^{bravo}$ and $q_B^{wox5}$ have been assigned. To this end, we first write down the following ratios:

$$\frac{F^{bravo}}{F^{WT}} = \frac{1 + T_B^{WT} + T_W^{WT} + T_{BW}^{WT}}{1 + T_W^{WT} q_W^{bravo}} = \frac{b}{c}$$

$$\frac{F^{wox5}}{F^{WT}} = \frac{1 + T_B^{WT} + T_W^{WT} + T_{BW}^{WT}}{1 + T_B^{WT} q_B^{wox5}} = \frac{c}{a}$$

$$\frac{F^{dm}}{F^{WT}} = 1 + T_B^{WT} + T_W^{WT} + T_{BW}^{WT} = \frac{d}{a}$$

and then isolate each contribution term, such that the following is found:

$$T_B^{WT} \pm \delta T_B^{WT} = \frac{1}{q_B^{wox5}}\left(\frac{d}{c} - 1\right) \pm \frac{1}{q_B^{wox5}}\sqrt{\left(\frac{\delta d}{c}\right)^2 + \left(\frac{d}{c^2}\delta c\right)^2}$$

$$T_W^{WT} \pm \delta T_W^{WT} = \frac{1}{q_W^{wox5}}\left(\frac{d}{b} - 1\right) \pm \frac{1}{q_W^{wox5}}\sqrt{\left(\frac{\delta d}{b}\right)^2 + \left(\frac{d}{b^2}\delta b\right)^2}$$

$$T_{BW}^{WT} \pm \delta T_{BW}^{WT} = \frac{d}{a} - 1 - \frac{1}{q_B^{wox5}}\left(\frac{d}{c} - 1\right) - \frac{1}{q_W^{bravo}}\left(\frac{d}{b} - 1\right)$$
$$\pm \sqrt{\left(\delta d\left(\frac{1}{a} - \frac{1}{q_B^{wox5}c} - \frac{1}{q_W^{bravo}b}\right)\right)^2 + \left(\frac{d}{a^2}\delta a\right)^2 + \left(\frac{d}{q_W^{bravo}b^2}\delta b\right)^2 + \left(\frac{d}{q_B^{bravo}c^2}\delta c\right)^2}$$

where the errors have been estimated using error propagation of the errors in $a,b,c$ and $d$ and assuming their independency. These equations enable to compute the BRAVO-mediated, WOX5-mediated and jointly BRAVO and WOX5-mediated contribution to QC divisions in the WT from the empirical data on QC divisions ($a,b,c,d$ values) and from $q_W^{bravo}$ and $q_B^{wox5}$ values. In Fig 5, continuous lines correspond to the best estimated values (e.g. $T_B^{WT} = \frac{1}{q_B^{wox5}}\left(\frac{d}{c} - 1\right)$), and the shaded area represents the range within the errors (e.g. $T_B^{WT} \pm \delta T_B^{WT}$).

We assigned values to $q_W^{bravo}$ and $q_B^{wox5}$ as follows. We first estimated their values through the fold-changes in promoter expression in the mutants. The fold-change of *pWOX5:GFP* expression in the *bravo* mutant compared to the WT is approximately 0.8 (Fig 2I), and this suggests $q_W^{bravo} = 0.8$. Hence, BRAVO slightly increases the WOX5-mediated contribution to QC division. The fact that *wox5* mutant exhibits phenotypes that have not been observed in the *bravo* mutant, such as CSC differentiation, also suggests that $q_W^{bravo}$ is not too small. For the effect that WOX5 has on BRAVO-mediated contribution, the estimate for $q_B^{wox5}$ based on the fold-change of *pBRAVO:GFP* expression in the *wox5* mutant is $q_B^{wox5} = 0.5$ (Fig 2H). Yet, from the root phenotypes of the mutants we cannot exclude other values. This is also supported by the results of the complex formation model (Appendix Fig S10). The results of this model

indicated that the amount of BRAVO protein not bound to WOX5 could decrease in the *wox5* mutant when there is a competing factor that binds BRAVO and WOX5 separately (Appendix Fig S10F). In this mutant, the competing factor would sequester BRAVO more than in the WT. Hence, smaller $q_B^{wox5}$ values and hence stronger upregulation of BRAVO-mediated contribution to QC division by WOX5 could be expected than that predicted from fold-changes in promoter expression. In contrast, the complex formation model predicts that the amount of WOX5 protein not bound to BRAVO does not decrease much in the *bravo* mutant, and hence, the model supports that $q_W^{bravo}$ is not small (Appendix Fig S10G). Therefore, we set $q_W^{bravo} = 0.8$ and evaluated the values of the contributions to QC divisions for different values of $q_B^{wox5}$.

## Data availability

The datasets and computer code produced in this study are available in the following databases:

- RNA-seq data: Gene Expression Omnibus GSE173945 (https://www.ncbi.nlm.nih.gov/geo/query/acc.cgi?acc=GSE173945).
- Modelling computer scripts: GitHub (https://github.com/josepmercadal/phd/tree/master/BRAVO_WOX5).
- RNA-seq data analysis script: GitHub (https://github.com/IsabelBetegon/BravoWox5_Manuscript).

**Expanded View** for this article is available online.

## Acknowledgements

A.I.C-D. is a recipient of a BIO2016-78955 grant from the Spanish Ministry of Economy and Competitiveness and a European Research Council, ERC Consolidator Grant (ERC-2015-CoG – 683163). I.B-P. is funded by the FPU15/02822 grant from the Spanish Ministry of Education, Culture and Sport; N.B. by the FI-DGR 2016FI_B 00472 grant from the AGAUR, Generalitat de Catalunya; and A.P-R. by the SEV-2015-0533 from the Severo Ochoa Programme for Centers of Excellence in R&D. M.I. and J.M. acknowledge support from the Spanish Ministry of Economy and Competitiveness and FEDER (EU) through grant FIS2015-66503-C3-3-P, from Ministerio de Ciencia, Innovación y Universidades / Agencia Estatal de Investigación / Fondo Europeo de Desarrollo Regional, Unión Europea through grant PGC2018-101896-B-I00 and from the Generalitat de Catalunya through Grup de Recerca Consolidat 2014 SGR 878 and 2017 SGR 1061. J.M. is funded by the Spanish Ministry of Education through BES-2016-078218. Y.S. and R.C.D. were funded by the Deutsche Forschungsgesellschaft (DFG) (grant STA12/12 1-1). A.I.C.-D.–A.C. collaboration was funded by the European Regional Development Funds and Marie Curie IRSES Project DEANN (PIRSES-GA-2013-612583). CRAG is funded by "Severo Ochoa Programme" from Centers of Excellence in R&D 2016-2019 (SEV-2015-485 0533).

## Author contributions

AIC-D and MI designed and supervised the study. IB-P, NB, AP-R, MM-B and JV-B and performed the experiments. JM, DF and MI formulated the mathematical modelling. JM performed the numerical simulations of the mathematical models. YS and RCB performed and analysed the FRET-FLIM assays. SP and CM collaborated in the Y2H and BiFC assays. IB-P performed QC-specific transcriptomics experiments and data analysis in collaboration with RS and AC. IB-P, JM, MI and AIC-D wrote the manuscript and all authors revised the manuscript.

## Conflict of interest

The authors declare that they have no conflict of interest.

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
