## [Review Process File · Molecular Systems Biology]

Precise transcriptional control of cellular quiescence by BRAVO/WOX5 complex in Arabidopsis roots

Isabel Betegón-Putze, Josep Mercadal, Nadja Bosch, Ainoa Planas-Riverola, Mar Marquès-Bueno, Josep Vilarrasa-Blasi, David Frigola, Rebecca Burkart, Cristina Martinez, Yvonne Stahl, Salome Prat, Ana Conesa, Rosangela Sozzani, Marta Ibanes, and Ana Caño-Delgado

DOI: 10.15252/msb.20209864

Corresponding author(s): Ana Caño-Delgado (ana.cano@cragenomica.es) , Marta Ibanes (miban@ub.edu)

Review Timeline:

Submission Date:	14th Jul 20
Editorial Decision:	27th Aug 20
Revision Received:	12th Mar 21
Editorial Decision:	8th Apr 21
Revision Received:	7th May 21
Accepted:	10th May 21

Editor: Maria Polychronidou

Transaction Report:

Thank you again for submitting your work to Molecular Systems Biology. We have now heard back from the three referees who agreed to evaluate your study. As you will see below, the reviewers acknowledge that the study is interesting and well done. However, they raise some concerns, which we would ask you to address in a revision.

Without repeating all the points listed below, some of the more fundamental issues are the following:

- As reviewers #2 and #3 mention, the transcriptional regulatory role of the WOX5/ BRAVO complex needs to be better supported. Some experimental evidence supporting the role of the complex in controlling QC division would also enhance the impact of the work.
- Providing evidence for the WOX5/BRAVO interaction in the root stem cell niche would significantly strengthen the reported conclusions.
- Reviewer #3 recommends some experimental validations for the hypothesis that bravo alleviates wox5 inhibition.
- Reviewer #2 suggests experimentally testing the hypothesis of possible competition between BRAVO/WOX5 for BES1/TPL. While we agree that this is a very good suggestion, we do not think that these experiments are mandatory for the acceptance of the work. That said, we would not be opposed to their inclusion should you feel inclined to perform them or in case you already have such data at hand.

All other issues raised by the referees would need to be satisfactorily addressed. Please let me know in case you would like to discuss in further detail any of the issues raised.

On a more editorial level, we would ask you to address the following issue.

Reviewer #1:

Betegón-Putze et al. Precise transcriptional control of cellular quiescence by BRAVO/WOX5 complex in Arabidopsis roots

Like animals, plant organs employ organising centres to control their stem cell niches. The molecular regulation of organising centres is a topic of great interest to both plant and animal researchers. The current manuscript addresses the regulatory relationship between 2 key proteins BRAVO and WOX5 whose interplay functions to repress division of organising centre cells (termed the QC in plants).

The authors start by reporting that mutants lacking either or both BRAVO and WOX5 gene products exhibit the same increased level of cell division in root QC, inferring they function interdependently.

They also report a reduction in their lateral root (LR) density. NOTE: it is not clear how this LR phenotype relates to loss of BRAVO and WOX5 expression in the QC of primary roots? Does loss of their gene products impact activity in the root basal meristem from which LRs initiate? Or is this a result of the loss of BRAVO and WOX5 expression in the QC of new LRs? This LR information is not strictly required for the paper and its central message so I would recommend removing it to avoid reader confusion.

Next, the authors explore how BRAVO and WOX5 interact - describing (by using transcriptional reporters) that BRAVO and WOX5 expression is co-localized in QC cells - careful measurements in mutant and inducible backgrounds revealed that WOX5 restricts its own expression to the QC, while BRAVO-dependent activation of WOX5 acts upstream of its autoregulation. NOTE: I was therefore confused by the title of this sub-section? - 'BRAVO and WOX5 reinforce each other at the root stem cell niche.' Equally confusing was the inclusion of data about brassinosteroids which has no impact on BRAVO and WOX5. I recommend removing this last paragraph to avoid reader confusion.

To probe the regulatory relationship between BRAVO and WOX5, the authors turned to modelling combined with molecular cell biology-based validation. Modelling revealed WOX5 induces BRAVO, which then goes on to alleviate WOX5 self-inhibition. Intriguingly, FRET-FLIM and Y2H assays revealed WOX5 and BRAVO directly interact to form a heterodimeric transcriptional complex. Further modelling analyses highlighted the importance of the BRAVO/WOX5 heterodimeric complex controlling QC divisions, helping explain the single and double mutant phenotypes.

Overall, this is a nice manuscript within the remit of MSB which utilises an elegant systems-based approach to help unravel the relationship between 2 key regulators of stem cell niche activity in plant roots.

Reviewer #2:

Betegón-Putz et al. report on the joint contribution of WOX5 and BRAVO transcription factors of cell division in the quiescent centre (QC) of the Arabidopsis root. Using genetic analysis they show that WOX5 & BRAVO have cross contribution to their respective

expression and quiescence of the QC. They establish a transcriptional regulation network which they numerically analyse. They also show that WOX5 & BRAVO can interact in the nucleus using Y2H, FRET-FLIM and splitYFP (the latter two in tobacco). They numerically analyse the contribution of a joint action of WOX5+BRAVO to QC quiescence by building a second model. Together their data support that WOX5 & BRAVO cross regulation at the transcriptional level and as a complex ensure QC division homeostasis.

Overall it is a very good manuscript, addressing an interesting question in an appropriate manner and which conclusions are well supported by the data presented. It is also very well written and the figures and methods are very good.

I have several major and minor recommendations.

Major points:

- the transcriptional regulation network (Fig 3) postulates that BRAVO alleviates WOX5 auto-repression. Although a likely scenario, it is not necessarily the only one in light of the effect of bravo on pWOX5 expression. Have the authors considered that BRAVO + X activates WOX5 and independently WOX5 self represses? If not, they should consider the impact of such wiring on the behaviour of the whole GRN.
- The likely co-existence of WOX5 & BRAVO in a protein complex has implications that should be explored:
 - the authors mention but do not test the hypothesis of possible competition between BRAVO/WOX5 for BES1/TPL. Y2H and/or FRET-FLIM measurements in presence of untagged BRAVO/WOX5 could be performed
 - the evidence for WOX5/BRAVO interaction are done in ectopic systems (yeast or OX in tobacco). Providing evidences for existence of the complex (at physiological levels of expression) in the SCN (e.g. co-IP from roots) would significantly strengthen the message.
 - what's the impact of the protein/protein interaction on the behaviour of the transcriptional GRN?

I leave it to the editor to decide whether addressing all or part of these is required for acceptance.

Minor points:

- l82: remove "and adaptation to the environment" as this aspect is not at all touched in the ms.
- Figure 1G, and similar panels in S2-S4: significance grouping is shown with letters and it is mentioned that a t-test was used. More details should be mentioned (ANOVA?, which post-hoc test exactly? which correction for multiple testing?)
- Figure 2 / S3/ S4 / S5 / S7: the scale bars are barely visible. Use colour blind friendly combination (e.g. magenta / green).
- l138-139: the expansion of pWOX5::GFP is not only toward the provascular cells, also toward the CSC.
- Figure 3C: clarify the meaning of the dagger exponent in the legend
- l190: remove "directly" as none of the technique shows direct interaction, rather co-occurrence in a complex.
- l239: replace " $q_{\{B\}^{\{Wm\}}$ being ..." with "with $q_{\{B\}^{\{Wm\}}$ being ..."
- Figure 5A legend: there is a typo on line 6, subscript & exponent are swapped. The sentence "while the individual contribution of BRAVO only increases for small values of q_{BWm} ." is unclear - I

see no increase. Reformulate.

Reviewer #3:

In this work the authors propose a regulatory circuit of BRAVO and WOX5 involved in the control of cellular quiescence. The connections and relations in the circuit are based on the analysis of transcriptional reporters in single and double mutants, and some qPCR in gain of function. Mathematical modelling is then applied to predict which factor will affect most the quiescence, bravo, wox or the complex. Conceptually this is an exciting approach but there are a few major assumptions which I think are untested, and the outcome of the prediction of the model is also not experimentally confirmed/tested. The understanding of these regulatory circuits is the next challenge in plant biology, and this approach offers significant contributions.

1) The hypothesis that bravo alleviates wox5 inhibition seems not really tested. Given that wox5 mutant has low levels of bravo transcription, it is equivalent to the bravo wox double in the case of wox five expression and it does not really inform on an alleviation. This could be tested in an uncoupled circuit by reintroduction of bravo expression under control of a different-inducible-promotor in the double mutant which carries the wox::GFP reporter. The observation that wox5 is expressed at lower levels in the bravo mutant can be linked to direct stimulation (limited by the presence of another factor) of Wox5 by bravo, and if bravo is alleviating wox 5 auto inhibition would we not expect a much stronger wox 5 signal in induced 35:BRAVO-E ?

2) It is well demonstrated that WOX5 and BRAVO can form a complex here, but how do we know that this complex mediates transcription, what is intended by transcriptional complex.

3) related to this, the prediction of the model is that the complex has a strong contribution to the control of QC divisions but this is not experimentally confirmed.

other concerns:

-turning to modelling instead of turning into modelling

-line 55, I believe VSC and CSC have the same proximity/distance to the qc

-103 lower LR density

-I struggle a bit about how GFP signals are measured and reported in the graphs, as an entire area is integrated an increase can be caused to more cells expressing and/or higher levels in a single cell.

Related to this in Figure 2. What is unit on fig 2H Y axis, left panel, average intensity of a pixel in millions, isn't this a value between 0-255 for a 8 bit measurement ? Or is this the integrated fluorescence of the entire area ? I also struggle with the title of this figure, as I see little evidence of reinforcement. Can the authors include bravo::GFP in the double mutant ? The line is reported in support materials, but it would be good to have a high quality acquisition here as well

-140 autoregulation, it is also possible that in wox5 mutants other signals reach neighbouring cells in absence of a functional QC. Has binding of WOX5 to pWOX5 been demonstrated ?

-201: What is the underpinning evidence for the statements on affinity ? Have bound and unbound fractions been quantified ?

=to link the model to the empirical evidence, why are median values used rather than average values. Can other possibilities be tested as well by the modelling approach, for example bravo and wox sequester each other but the complex has no regulatory contribution

Point-by-point reviewer response**REVIEWER #1**

Betegón-Putze et al. Precise transcriptional control of cellular quiescence by BRAVO/WOX5 complex in Arabidopsis roots

Like animals, plant organs employ organising centres to control their stem cell niches. The molecular regulation of organising centres is a topic of great interest to both plant and animal researchers. The current manuscript addresses the regulatory relationship between 2 key proteins BRAVO and WOX5 whose interplay functions to repress division of organising centre cells (termed the QC in plants). The authors start by reporting that mutants lacking either or both BRAVO and WOX5 gene products exhibit the same increased level of cell division in root QC, inferring they function interdependently.

They also report a reduction in their lateral root (LR) density. NOTE: it is not clear how this LR phenotype relates to loss of BRAVO and WOX5 expression in the QC of primary roots? Does loss of their gene products impact activity in the root basal meristem from which LRs initiate? Or is this a result of the loss of BRAVO and WOX5 expression in the QC of new LRs? This LR information is not strictly required for the paper and its central message so I would recommend removing it to avoid reader confusion.

ANSWER: The ms. has been rewritten and reorganized to fit the new data during this revision, so now in agreement with this reviewer request we have downplayed our LR data in the ms.

Next, the authors explore how BRAVO and WOX5 interact - describing (by using transcriptional reporters) that BRAVO and WOX5 expression is co-localized in QC cells - careful measurements in mutant and inducible backgrounds revealed that WOX5 restricts its own expression to the QC, while BRAVO-dependent activation of WOX5 acts upstream of its autoregulation. NOTE: I was therefore confused by the title of this sub-section? - 'BRAVO and WOX5 reinforce each other at the root stem cell niche.' Equally confusing was the inclusion of data about brassinosteroids which has no impact on BRAVO and WOX5. I recommend removing this last paragraph to avoid reader confusion.

ANSWER: In the revised version of the ms. we have removed it as suggested by the reviewer. We have maintained the supp. figure this paragraph was referring to since the results of the new mathematical model in New Figure 3 referred to it.

To probe the regulatory relationship between BRAVO and WOX5, the authors turned to modelling combined with molecular cell biology-based validation. Modelling revealed WOX5 induces BRAVO, which then goes on to alleviate WOX5 self-inhibition. Intriguingly, FRET-FLIM and Y2H assays revealed WOX5 and BRAVO directly interact to form a heterodimeric transcriptional complex. Further modelling analyses highlighted the importance of the BRAVO/WOX5 heterodimeric complex controlling QC divisions, helping explain the single and double mutant phenotypes.

Overall, this is a nice manuscript within the remit of MSB which utilises an elegant systems-based approach to help unravel the relationship between 2 key regulators of stem cell niche activity in plant roots.

ANSWER: We appreciate the positive evaluation of our manuscript by this reviewer and we believe that the revised version of the ms. (that has taken unexpectedly longer due to the pandemic lockdowns) is definitively all-embracing now.

REVIEWER #2

Betegón-Putz et al. report on the joint contribution of WOX5 and BRAVO transcription factors of cell division in the quiescent centre (QC) of the Arabidopsis root.

Using genetic analysis they show that WOX5 & BRAVO have cross contribution to their respective expression and quiescence of the QC. They establish a transcriptional regulation network which they numerically analyse. They also show that WOX5 & BRAVO can interact in the nucleus using Y2H, FRET-FLIM and splitYFP (the latter two in tobacco). They numerically analyse the contribution of a joint action of WOX5+BRAVO to QC quiescence by building a second model. Together their data support that WOX5 & BRAVO cross regulation at the transcriptional level and as a complex ensure QC division homeostasis.

Overall it is a very good manuscript, addressing an interesting question in an appropriate manner and which conclusions are well supported by the data presented. It is also very well written and the figures and methods are very good.

ANSWER: We do appreciate the positive evaluation and the constructive suggestions on our future work.

I have several major and minor recommendations.

Major points:

- the transcriptional regulation network (Fig 3) postulates that BRAVO alleviates WOX5 auto-repression. Although a likely scenario, it is not necessarily the only one in light of the effect of bravo on pWOX5 expression. Have the authors considered that BRAVO + X activates WOX5 and independently WOX5 self represses? If not, they should consider the impact of such wiring on the behaviour of the whole GRN.

ANSWER: We thank the reviewer for the suggestion. We have now modelled the proposed scenario. As expected by the reviewer, this model is able to reproduce the data as well. Therefore, we now show two scenarios (the alleviation model and the activation model) that can explain the data. We have also analysed the sensitivity of both models to parameter variation and found that the alleviation model is more robust to reproduce the data. In particular, the activation model can fail to reproduce the average increase of WOX5 expression in the *bravo wox5* mutant, and in the *wox5* mutant, while the alleviation model cannot.

We have included the new model and results concerning both models in the revised version of the manuscript:

- **The section presenting the mathematical modeling results of the cross-regulations (now named “Two possible scenarios for BRAVO-WOX5 cross-regulations”) has been**

- modified to present the new model and its results, together with the alleviation model.
- Figure 3 has been changed. New Figure 3 presents the results of both models. It includes a parameter space exploration of the results (panels E,F). Because of space, previous results showing how the model solutions reach stationarity have been removed (former panel B).
 - Two new Supporting Figures have been added (Fig. S8, S9), which analyse both models.
 - Figure S8 shows the results of promoter activities mimicking each plant. This has been added to facilitate comparison between modeling data and experimental data.
 - Figure S9 shows results of large space parameter exploration (complementing the new exploration reported now in Figure 3E,F).
 - Methods concerning the mathematical model and Supporting Information Text have been revised to include the new model.

To facilitate the comparison between the formulations of the two models and to reproduce the overexpression experiments, besides those in mutants, some new parameter values have been chosen for the alleviation model (included in revised Table S1). For the parameter space exploration we have performed non-dimensional formulation of the alleviation and activation models and have worked with them.

- The likely co-existence of WOX5 & BRAVO in a protein complex has implications that should be explored:
 - the authors mention but do not test the hypothesis of possible competition between BRAVO/WOX5 for BES1/TPL. Y2H and/or FRET-FLIM measurements in presence of untagged BRAVO/WOX5 could be performed.
 - the evidence for WOX5/BRAVO interaction are done in ectopic systems (yeast or OX in tobacco). Providing evidences for existence of the complex (at physiological levels of expression) in the SCN (e.g. co-IP from roots) would significantly strengthen the message.

ANSWER: We carried IP the BRAVO complex from roots. Indeed, we have even run MS-analysis in U. Wageningen to IP BRAVO interactors from root, similar to what we previously successfully did at Fabregas et al., Plant Cell 2014. Unfortunately, the reduced amount of native BRAVO and their specific localization to a few cells of the root apex hampered the purification of interactors by mass spec. In the original ms. we provided experimental evidence using in vitro (Y2H) and in planta (BIFC) different methods that demonstrate the biochemical interaction of BRAVO/WOX5. Importantly, these findings are further sustained by the new mathematical modeling provided in the revised version of the ms., all highlighting the significance of system biological approaches to understand complex regulatory circuits as those of plant stem cells.

- what's the impact of the protein/protein interaction on the behaviour of the transcriptional GRN?

ANSWER: We have now formulated a new model that includes the BRAVO-WOX5 protein interaction and the protein interaction of BRAVO and of WOX5 with another protein (named S in the model, for sequestering competing factor). This protein can be TPL and/or BES1. The new model considers that BRAVO is transcriptionally activated by WOX5 protein. It also considers that WOX5 and BRAVO each transcriptionally repress their own expression (notice that these are the common regulations of the alleviation and the activation models). Our results show that the impact of the BRAVO/WOX5 protein interaction is to drive the alleviation mechanism. Since BRAVO binds to WOX5, BRAVO sequesters partially WOX5 and impedes WOX5 from self-repressing as much as it could. Therefore, by sequestering WOX5 in the BRAVO-WOX5 complex, BRAVO can alleviate WOX5 self-repression.

Through the model, we have also addressed the competition of BRAVO and WOX5 for another factor (S), such as TPL or BES1D. Our numerical results show that the model that considers all these protein/protein interactions is able to reproduce the trends of the data. Yet, the presence of this other factor, if being very abundant for instance, can impede BRAVO from regulating WOX5 and vice versa. However, by being able to interact with WOX5 and to interact with BRAVO, the factor S becomes less detrimental for the cross-regulations between BRAVO and WOX5. Thus, BRAVO-WOX5 cross-regulations become less sensitive to the presence of the competing factor when it can bind both BRAVO and WOX5 separately. All these results are presented in a new section entitled "The BRAVO-WOX5 complex provides a mechanism for the alleviation model".

Another interesting result obtained by considering the competing factor S is the response that the amount of protein not bound to BRAVO or WOX5 has in the mutants. Our results show that when the competing factor S is included, together with the BRAVO/WOX5 protein/protein interaction, then the changes observed in expression can correlate with and even reinforce the changes in the amount of available (not sequestered) protein in the mutants. In other words, the changes found for the expression in the single mutants translate and even become increased in the amount of available protein, if the competing factor is present. This does not happen if the competing factor is not present. These new results have been added in the section where the contribution of the complex to regulate QC divisions is presented (section now named "BRAVO-WOX5 complex is sufficient for the control of QC divisions") since they help to define the expected range of $q_{B^{Wm}}$ and $q_{W^{Bm}}$ values. In Methods, subsection "A model to compute the contribution of BRAVO and WOX5 to regulate QC division" we have deleted the information on how to estimate $q_{B^{Wm}}$ and $q_{W^{Bm}}$ since it is now presented in the Results.

All these new results are presented in new Figure S10. The new model is explained now in Methods and Supporting Information Text. Table S1 now includes the parameter values of this new model.

I leave it to the editor to decide whether addressing all or part of these is required for acceptance.

Minor points:

- l82: remove "and adaptation to the environment" as this aspect is not at all touched in the ms.

ANSWER: done.

- Figure 1G, and similar panels in S2-S4: significance grouping is shown with letters and it is mentioned that a t-test was used. More details should be mentioned (ANOVA?, which post-hoc test exactly? which correction for multiple testing?)

ANSWER: We used the Student's t test and the different letters indicate statistically significant differences using p-value < 0.05.

- Figure 2 / S3/ S4 / S5 / S7: the scale bars are barely visible. Use colour blind friendly combination (e.g. magenta / green).

ANSWER: Yes, we now use colour blind friendly combination in the figures of the revised version of the ms.

- l138-139: the expansion of pWOX5::GFP is not only toward the provascular cells, also toward the CSC.

ANSWER: We added this observation in the revised ms. text.

- Figure 3C: clarify the meaning of the dagger exponent in the legend

ANSWER: The meaning is now defined in the new Figure legend.

- l190: remove "directly" as none of the technique shows direct interaction, rather co-occurrence in a complex.

ANSWER: This has been amended.

- l239: replace " q_{B}^{Wm} being ..." with "with q_{B}^{Wm} being ..."

ANSWER: This has been amended.

- Figure 5A legend: there is a typo on line 6, subscript & exponent are swapped. The sentence "while the individual contribution of BRAVO only increases for small values of q_{BWm} ." is unclear
- I see no increase. Reformulate.

ANSWER: We do not find the swapped subscript & exponent. The sentence has been changed to "The individual contribution of BRAVO becomes significant only for small values of q_{B}^{Wm} ."

REVIEWER #3:

In this work the authors propose a regulatory circuit of BRAVO and WOX5 involved in the control of cellular quiescence. The connections and relations in the circuit are based on the analysis of transcriptional reporters in single and double mutants, and some qPCR in gain of function. Mathematical modelling is then applied to predict which factor will affect most the quiescence, bravo, wox or the complex. Conceptually this is an exciting approach but there are a few major assumptions which i think are untested, and the outcome of the prediction of the model is also not experimentally confirmed/tested. The understanding of these regulatory circuits is the next challenge in plant biology, and this approach offer significant contributions.

ANSWER: We appreciate the positive evaluation on our manuscript. Indeed, we really acknowledge the very insightful suggestions as some of the open new avenues of research

that will certainly let us continue in the deciphering of the understanding of gene regulatory circuits that operate during stem cell development in plants.

1)The hypothesis that bravo alleviates wox5 inhibition seems not really tested. Given that wox5 mutant has low levels of bravo transcription, it is equivalent to the bravo wox double in the case of wox five expression and it does not really inform on an alleviation. This could be tested in an uncoupled circuit by reintroduction of bravo expression under control of a different-inducible-promotor in the double mutant which carries the wox::GFP reporter. The observation that wox5 is expressed at lower levels in the bravo mutant can be linked to direct stimulation (limited by the presence of another factor) of Wox5 by bravo, and if bravo is alleviating wox 5 auto inhibition would we not expect a much stronger wox 5 signal in induced 35:BRAVO-E ?

ANSWER: We agree that the proposed experiment can provide support for the alleviation hypothesis. Unfortunately, these experiments would take much longer than desired because we do not have the required lines. To address the concern raised by the reviewer we took a modeling approach. We now show through a new mathematical model that the formation of the BRAVO/WOX5 complex can mediate the alleviation hypothesis (see previous answer to Reviewer 2). Thus, albeit we do not test the alleviation mechanism, we now suggest it is a natural mechanism to occur when the BRAVO-WOX5 protein complex is considered. We also now model the stimulation hypothesis (activation model) and show it is consistent with the empirical data. To compare the alleviation model with the activation model we have performed random parameter variation and evaluate the outcomes of each model. The results show that the activation model is less robust than the alleviation model. See also previous answer to Reviewer 2 on this model.

In addition, we have now modelled the overexpression experiments in all models and show the agreement of the corresponding results with the empirical data. The results are depicted in new Figs.3, S8-S10. Parameter values of the alleviation model have been changed to fit all the data and be comparable between models. Methods describing the modeling of overexpression have been added in Methods.

As indicated by the reviewer, if BRAVO is alleviating WOX5 auto-inhibition, it can be expected that WOX5 increases in 35S:BRAVO-E. The same expectation can be raised if BRAVO stimulates WOX5. In both scenarios, a limiting factor can be postulated to impede additional action of BRAVO, or it can be assumed that BRAVO action is at its maximal capabilities already in the WT, such that BRAVO overexpression does not drive an increased WOX5 expression. In the two models, parameters B_0 and W_0 can control the extent of the WOX5 increase in the in silico overexpression experiment and be chosen such that this increase is very small. Therefore, the expected outcomes of the stimulation and the alleviation scenario are similar under overexpression of BRAVO and can be resolved similarly to match the data.

For the new model which takes into account the BRAVO-WOX5 protein complex formation, we find that the presence of additional sequestering factors, which are suggested by our data on the binding of BRAVO and of WOX5 to TPL and BES1D, can act as limiting factors for the effect of BRAVO mediated alleviation upon BRAVO overexpression. This is now shown in new Figure S10 (when comparing panels A and C).

For a detailed description of the changes been made, see previous answer to Reviewer 2.

2) It is well demonstrated that WOX5 and BRAVO can form a complex here, but how do we know that this complex mediates transcription, what is intended by transcriptional complex.

ANSWER: We appreciate this insightful suggestion. To address the transcriptional regulatory role of the WOX5/ BRAVO complex we provide experimental evidence on the transcriptional role of BRAVO and WOX5 in the QC by doing RNAseq analysis coupled to FACS (fluorescent activated cell sorting) of stem cells in WT compared to that bravo and wox5 mutants. (in response to Reviewer #2 and Reviewer #3). These experiments have been carried in collaboration with Prof. Ross Sozzani (NCSU, Raleigh, USA) and Prof. Ana Conesa (U. Florida, USA), that now become co-authors in this study. We truly believe that these results enhance the impact of the work and represent the starting point to several other research groups, thus enhancing the citation of the article. These new results are shown in a new Figure 6 of the main ms. and reported in new section of the ms. entitled "BRAVO and WOX5 mediated transcriptional control in the QC" (pg.13 starting line 315).

Additionally, we have removed the word "transcriptional" from "transcriptional complex".

3) related to this, the prediction of the model is that the complex has a strong contribution to the control of QC divisions but this is not experimentally confirmed.

ANSWER: We agree with the reviewer that this contribution is not experimentally confirmed. Proper confirmation of it would require finding the domain where these two proteins interact and mutate it such that the proteins remain functional but unable to bind to each other. This confirmation, albeit extremely interesting, remains out of the scope of the current manuscript. To emphasize that the strong contribution of the complex is not experimentally confirmed by the current data but it is only predicted, we have rephrased partially the section of "BRAVO-WOX5 complex is relevant for the control of QC divisions", which has been changed to "BRAVO-WOX5 complex is sufficient for the control of QC divisions". We have also rephrased the Discussion. We now indicate that the complex is sufficient (instead of necessary) to explain the QC division data. In the revised manuscript we also describe in more detail additional, overlapping, scenarios which cannot be discarded, such as BRAVO being downstream WOX5 to repress QC divisions. Moreover, we now explicitly discuss that our results suggest that WOX5 does not regulate QC divisions without BRAVO. We aim that the new exposition and discussion of results can help advance in this field in the future.

other concerns:

-turning to modelling instead of turning into modelling

ANSWER: This has been corrected.

-line 55, I believe VSC and CSC have the same proximity/distance to the qc

ANSWER: Ok.

-103 lower LR density

ANSWER: This has been corrected.

-I struggle a bit about how GFP signals are measured and reported in the graphs, as an entire area is integrated an increase can be caused to more cells expressing and/or higher levels in a single cell. Related to this in Figure 2. What is unit on fig 2H Y axis, left panel, average intensity of a pixel in millions, isn't this a value between 0-255 for a 8 bit measurement ? Or is this the integrated fluorescence of the entire area ? I also struggle with the title of this figure, as I see little evidence of reinforcement. Can the authors include *bravo::GFP* in the double mutant ? The line is reported in support materials, but it would be good to have a high quality acquisition here as well

ANSWER:

The GFP intensity was quantified with ImageJ using the Integrated Density value, so it corresponds to the integrated fluorescence of the entire area. We maintained the same settings for all the images for comparisons. We already include the high-quality acquisition images of the *pBRAVO:GFP;bravo wox5* line. They are shown in Figure S4. They are shown in a different figure because the laser settings were different due to the renovation of the lasers in the confocal microscope.

-140 autoregulation, it is also possible that in *wox5* mutants other signals reach neighbouring cells in absence of a functional QC. Has binding of WOX5 to pWOX5 been demonstrated ?

ANSWER: Not to our knowledge, but we are not assuming this to happen.

-201:What is the underpinning evidence for the statements on affinity ? Have bound and unbound fractions been quantified?

ANSWER: We did not quantify the bound/unbound fractions, which would be rather difficult in these complex in vivo plant samples regarding different levels of expected autofluorescence. Therefore we looked at the donor lifetime and consistently got even further decreased lifetimes of the BRAVO and WOX5 donors in the case of BES1D co-expressed as an acceptor. We have seen this effect in at least three independent replicate measurements. This is why we argue that the interaction of BES1D is stronger in comparison to e.g. BES1. We analysed the data by statistical tests (one-way ANOVA with Sidakholm posthoc test, where differences are indicated by different letters, $p\text{-value} < 0,01$ and $n > 20$).

=to link the model to the empirical evidence, why are median values used rather than average values. Can other possibilities be tested as well by the modelling approach, for example *bravo* and *wox* sequester each other but the complex has no regulatory contribution

ANSWER: Now we use mean values. Very similar fold-changes are obtained with median and mean values. Error bars have been added computed through error propagation of each mean expression value. As detailed in the first response, we have added two new models which test two new possibilities: 1) the activation model, according to which BRAVO induces WOX5; and 2) That BRAVO and WOX5 sequester each other and the complex has no regulatory contribution on the regulation of BRAVO and WOX5 expressions. This model indicates that this sequestering process can drive the alleviation hypothesis proposed. See previous answers to Reviewer 2 for a description of these models and the changes been made.

Thank you again for sending us your revised manuscript. We have now heard back from reviewers #2 and #3 who were asked to evaluate your study. As you will see below, both reviewers think that the study has improved as a result of the performed revisions and they are supportive of publication. However, they still list a few remaining concerns, which we would ask you to address in a minor revision.

Moreover, we would ask you to address some remaining editorial issues listed below.

REFEREE REPORTS

Reviewer #2:

The authors have provided detailed responses to the points I had raised and amended the manuscript with several new data. The manuscript has gained from these additions. I still have some suggestion to further improve the manuscript:

- the writing could be condensed: in particular the section "BRAVO-WOX5 complex is sufficient for the control of QC divisions" and the discussion.
- L119-120 + Fig 2: the data related to pBRAVO::GFP in *wox5*/bravo (image + quantification) should be merged with the main figure.
- In Fig 2H: values of Y axis are missing.
- l146-168: it could be good to clarify this section by by stating up front that there are 2 models that differ by the way BRAVO regulates WOX5, then explain the model in details.
- Figure 3 & L183: The data related to BL are not absolutely necessary to the conclusion of the manuscript which is already quite dense. Maybe these could be removed completely or moved to supplemental material and then toned down in discussion?
- Figure 3/D: replace sigma by more biologically meaningful labels (eg. pBRAVO in bravo, *wox5* and bravo/*wox5*) to increase the clarity of the figure. I also wonder why the line for pWOX in bravo/*wox5* not visible whereas the experimental data point is shown.
- l186-189, Figure 3E,F, Figure S10 and in the rebuttal letter: although I appreciate the efforts of the authors to consider both Alleviation and Activation variant, it is not clear to me how they reach the conclusion that "the activation model can reproduce the trends of changes of WOX5 promoter expression in the mutants, especially in the bravo *wox5* mutant, in a larger parameter space". I can't see this clearly in the data. Could they clarify?
- l194-216 and Figure 4: Protein-protein interaction. I take note of the response of the authors about validating by co-IP the WOX/BRAVO interaction, which is acceptable; however as the interaction is an important element of the manuscript on which many interpretation rely, ascertaining the solidity of this is important. It only struck me reading the revised manuscript, but for all pair tested in FRET-FLIM, there is a reduction of lifetime which is taken as support for protein protein interaction. How much of this is biologically relevant? What would be the values for homodimers (e.g. BRAVO-mV + BRAVO-mCh) and for an artificial nuclear protein (e.g. BRAVO-mV + nls-mCh)? Ideally a mutant in either BRAVO or WOX5 that would suppress the interaction should

be tested.

- l275-6: "we considered that the individual contribution mediated by BRAVO is changed by a factor q_{BWm} in the *wox5* mutant compared to that in the WT (q_{BWm})." there is something broken in the sentence.

- l352-353: the claim that these TFs are possible interactors is based on what?

- l408-409: "the finding that BRAVO and WOX5 can bind together" replace "bind" by "interact"

Reviewer #3:

Tight regulation of cell division in quiescent cells is essential for the maintenance of the organising centre at the core of the stem cell niche in higher plants. In this manuscript Betegon-Putze et al provide and explore plausible models for the interaction between Bravo and WOX5 that mediate this quiescence in the stem cell niche of the root , mechanisms with enable the convergence of brassinosteroids and other signals on this process. The combination of imaging, mutant analysis and mathematical modelling offers an innovative approach to this fundamental biological problem, I very much welcome this revised version of the manuscript, and its additions provided. The exploration of the two models is insightful and the transcriptional regulation is valid.

A few minor editing comments which can be handled by the authors themselves

Fig 2H Y-axis label

can I suggest

"Average integrated intensity" to avoid confusion

Material and methods

Fret Flim

for clarity, upto the authors discretion, I suggest to add

As the distance between fluorophores in the fluorescent proteins limits the FRET efficiency, the lifetime can be used as an indicator for affinity of the tagged proteins

line 100 lower lateral root density

Below we answer point by point to the reviewer comments and suggestions:

Reviewer #2:

The authors have provided detailed responses to the points I had raised and amended the manuscript with several new data. The manuscript has gained from these additions.

I still have some suggestion to further improve the manuscript:

- the writing could be condensed: in particular the section "BRAVO-WOX5 complex is sufficient for the control of QC divisions" and the discussion.

ANSWER: The section "BRAVO-WOX5 complex is sufficient for the control of QC divisions" has been condensed by slightly re-writing it and by moving one paragraph to Material and Methods (subsection "A model to compute the contribution of BRAVO and WOX5 to QC divisions"). This subsection of Material and Methods has been partially re-written to accommodate this new paragraph.

The Discussion section has been condensed by partially re-writing in more synthesized manners some sentences and by removing paragraphs devoted to Brassinosteroid treatments.

- L119-120 + Fig 2: the data related to pBRAVO::GFP in *wox5*/*bravo* (image + quantification) should be merged with the main figure.

ANSWER: We can't merge them because the settings for obtaining the data were different

- In Fig 2H: values of Y axis are missing.

ANSWER: We added the values for the Y axis in Fig 2H.

- L146-168: it could be good to clarify this section by stating up front that there are 2 models that differ by the way BRAVO regulates WOX5, then explain the model in details.

ANSWER: Done it.

- Figure 3 & L183: The data related to BL are not absolutely necessary to the conclusion of the manuscript which is already quite dense. Maybe these could be removed completely or moved to supplemental material and then toned down in discussion?

ANSWER: Panels C and D in Figure 3, which contained information on BL besides information on CTL conditions, have been moved to Appendix Figure S8 (E,F). In addition, two cartoon panels have been added in this Appendix Figure, to clarify the variables used in panels E,F.

In the section where these panels were cited, the sentences devoted to BL have been slightly toned down. In the Discussion, we have removed all paragraphs referring to BL

- Figure 3/D: replace sigma by more biologically meaningful labels (eg. pBRAVO in *bravo*, *wox5*

and bravo/wox5) to increase the clarity of the figure. I also wonder why the line for pWOX in bravo/wox5 not visible whereas the experimental data point is shown.

ANSWER: The labels have been replaced to more meaningful labels that could fit within the space of the figures: e.g. pWbravo stands for WOX5 promoter in the bravo mutant. The labels in Figure 3 also appeared in Appendix Figures S8, S9 and S10, and hence all have been changed. The change has been also applied in Material and Methods, where the new labelling has been defined (substituting the former one), and in Appendix Text. Figure captions have been also updated for this change.

Since the new labelling involved a new nomenclature, which is more self-explanatory, to name the mutants within mathematical variables, we have also applied it to Figure 5 and to the variables related to the modeling of Figure 5 (in main text section "BRAVO-WOX5 complex is sufficient for the control of QC divisions") and in Material and Methods. Hence notation has been changed from qBWm to qBwox5, and analogously for the bravo mutant.

In Figure 3D, the line for pWOX5 in the double mutant overlaps that of the single wox5 mutant and this is why it is not noticeable. The explanation has been now explicitly indicated in the caption (Now in Appendix Figure S8).

- l186-189, Figure 3E,F, Figure S10 and in the rebuttal letter: although I appreciate the efforts of the authors to consider both Alleviation and Activation variant, it is not clear to me how they reach the conclusion that "the activation model can reproduce the trends of changes of WOX5 promoter expression in the mutants, especially in the bravo wox5 mutant, in a larger parameter space". I can't see this clearly in the data. Could they clarify?

ANSWER: The sentence indicated that "the alleviation model compared to the activation model can reproduce the trends of changes of WOX5 promoter expression in the mutants, especially in the bravo wox5 mutant, in a larger parameter space (Figures 3E, F, Appendix Figure S9)".

To clarify it we have changed it to:

"Exploration of the parameter space around the default parameter set (Appendix Table S1) indicated that both models can reproduce the trends of changes of WOX5 promoter expression in the mutants and in overexpression lines in a large parameter space (Figure 3C, D, Appendix Figure S9). Yet, the alleviation model performs better than the activation model. In larger parameter regions, this latter model can predict fold-changes that are opposite to those found in the experiments, especially in the bravo wox5 mutant (Figures 3C, D, Appendix Figure S9)."

For instance, in Figure 3F we see results for the bravo wox5 mutant in the activation model that correspond to fold-changes smaller than one (i.e. a decrease in the mutant), whereas in the experiments we have a fold-change larger than one.

To facilitate understanding, we indicate in the figure caption that the line of fold-change=1 is depicted to distinguish the regions of fold-change <1 and fold-change >1.

- I194-216 and Figure 4: Protein-protein interaction. I take note of the response of the authors about validating by co-IP the WOX/BRAVO interaction, which is acceptable; however as the interaction is an important element of the manuscript on which many interpretation rely, ascertaining the solidity of this is important. It only struck me reading the revised manuscript, but for all pair tested in FRET-FLIM, there is a reduction of lifetime which is taken as support for protein protein interaction. How much of this is biologically relevant? What would be the values for homodimers (e.g. BRAVO-mV + BRAVO-mCh) and for an artificial nuclear protein (e.g. BRAVO-mV + nls-mCh)? Ideally a mutant in either BRAVO or WOX5 that would suppress the interaction should be tested.

ANSWER: These are very challenging questions we try to address in our future work. Thanks.

- I275-6: "we considered that the individual contribution mediated by BRAVO is changed by a factor q_{BWm} in the *wox5* mutant compared to that in the WT (q_{BWm})." there is something broken in the sentence.

ANSWER: The sentence has been changed to "we assumed that the BRAVO-mediated contribution in the *wox5* mutant compared to the WT changes by a factor of $q_{B^{wox5}}$ ". Additional changes in this section has been made, as indicated in a response above, to condense the message.

- I352-353: the claim that these TFs are possible interactors is based on what?

ANSWER: We edited the discussion for clarification.

- I408-409: "the finding that BRAVO and WOX5 can bind together" replace "bind" by "interact"

ANSWER: Done it.

Reviewer #3:

Tight regulation of cell division in quiescent cells is essential for the maintenance of the organising centre at the core of the stem cell niche in higher plants. In this manuscript Betegon-Putze et al provide and explore plausible models for the interaction between Bravo and WOX5 that mediate this quiescence in the stem cell niche of the root, mechanisms with enable the convergence of brassinosteroids and other signals on this process. The combination of imaging, mutant analysis and mathematical modelling offers an innovative approach to this fundamental biological problem,

I very much welcome this revised version of the manuscript, and its additions provided. The exploration of the two models is insightful and the transcriptional regulation is valid.

A few minor editing comments which can be handled by the authors themselves

Fig 2H Y-axis label can I suggest "Average integrated intensity" to avoid confusion

ANSWER: We did not change the label in the axis. The method for quantification is described in the material and methods section.

Material and methods

Fret Flim

for clarity, upto the authors discretion, I suggest to add

As the distance between fluorophores in the fluorescent proteins limits the FRET efficiency, the lifetime can be used as an indicator for affinity of the tagged proteins

ANSWER: We edited the FRET FLIM method description for clarification.

line 100 lower lateral root density

ANSWER: Done it.

Thank you again for sending us your revised manuscript. We are now satisfied with the modifications made and I am pleased to inform you that your paper has been accepted for publication.

Corresponding Author Name: Ana I. Cano-Delgado
Journal Submitted to: Molecular Systems Biology
Manuscript Number: